# Robust tissue pattern formation by coupling morphogen signal and cell adhesion

Kosuke Mizuno[1,2,3], Tsuyoshi Hirashima [ID][4,5✉] & Satoshi Toda [ID][1,3✉]

## Abstract

**Morphogens, locally produced signaling molecules, form a concentration gradient to guide tissue patterning. Tissue patterns emerge as a collaboration between morphogen diffusion and responsive cell behaviors, but the mechanisms through which diffusing morphogens define precise spatial patterns amidst biological fluctuations remain unclear. To investigate how cells respond to diffusing proteins to generate tissue patterns, we develop SYMPLE3D, a 3D culture platform. By engineering gene expression responsive to artificial morphogens, we observe that coupling morphogen signals with cadherin-based adhesion is sufficient to convert a morphogen gradient into distinct tissue domains. Morphogen-induced cadherins gather activated cells into a single domain, removing ectopically activated cells. In addition, we reveal a switch-like induction of cadherin-mediated compaction and cell mixing, homogenizing activated cells within the morphogen gradient to form a uniformly activated domain with a sharp boundary. These findings highlight the cooperation between morphogen gradients and cell adhesion in robust tissue patterning and introduce a novel method for tissue engineering to develop new tissue domains in organoids.**

**Keywords** Pattern Formation; Synthetic Biology; Morphogen; Cell Adhesion; Cadherin
**Subject Categories** Cell Adhesion, Polarity & Cytoskeleton; Development; Signal Transduction

## Introduction

The spatial organization of cells in 3D tissue architecture is crucial for the generation of functional tissues. During embryogenesis, cell differentiation is spatially controlled by extracellular signaling molecules called morphogens. Morphogens are secreted by localized organizer cells, forming a concentration gradient across tissues through diffusion. Following these graded morphogen signals, receiver cells undergo variable transcriptional responses, leading to tissue patterning in position-specific cell types (Rogers and Schier, 2011).

A fundamental question regarding morphogen-based tissue patterning is how these gradients can robustly generate spatial patterns with distinct domains and sharp boundaries (Gurdon and Bourillot, 2001). In addition, natural gradients contain signal noise due to biological fluctuations, indicating that cells must interpret the morphogen signals amidst this noise (Akieda et al, 2019; Xiong et al, 2013). The molecular mechanisms underlying robust morphogen interpretation have been extensively studied using animal models such as Drosophila embryos and vertebrate neural tubes (Briscoe and Small, 2015). Previous studies have proposed several mechanisms contributing to tissue patterning, including gene regulatory networks, differential adhesion, juxtracrine signaling, and cell competition (Akieda et al, 2019; Balaskas et al, 2012; Dahmann et al, 2011; Tsai et al, 2020). While these mechanisms play important roles in various animal models, many types of cell behaviors and interactions occur in parallel in complex biological systems, making it challenging to determine the abilities of individual mechanisms for tissue patterning.

Recent advances in stem cell biology have enabled the development of 3D tissue culture systems called organoids, that mimic the cell composition and tissue morphology of their organ of origin. Similar to living tissues in animals, some organoids utilize morphogen gradients for development. For instance, intestinal and neural tube organoids rely on Wnt and Shh gradients, respectively, to generate organ architecture and cell components (Farin et al, 2016; Meinhardt et al, 2014). However, certain types of organoids, such as brain organoids, lack morphogen instructions, resulting in less cell diversity and spatial organization (Andrews and Kriegstein, 2022; Matsui et al, 2020). In current organoid culture protocols, biochemical cues, such as morphogens and niche factors, are supplemented into whole culture media, lacking the positional information provided by gradients observed during natural development. Therefore, reproducing morphogen gradients in vitro may improve the spatial organization of organoids (Cederquist et al, 2019; Hofer and Lutolf, 2021). Furthermore, the ability to control the spatial patterns of user-defined transcriptional programs is essential for engineering synthetic organoids (Trentesaux et al, 2023; Zheng and Loh, 2022), highlighting the need for a deeper understanding of the minimal components sufficient for morphogenic tissue patterning.

[1]WPI Nano Life Science Institute (NanoLSI), Kanazawa University, Kanazawa, Ishikawa, Japan. [2]Graduate School of Frontier Science Initiative, Kanazawa University, Kanazawa, Ishikawa, Japan. [3]Institute for Protein Research, Osaka University, Suita, Osaka, Japan. [4]Mechanobiology Institute, National University of Singapore, Singapore, Singapore. [5]Department of Physiology, Yong Loo Lin School of Medicine, National University of Singapore, Singapore, Singapore. ✉E-mail: thira@nus.edu.sg; satoshi.toda@protein.osaka-u.ac.jp

In this context, a synthetic biology approach would be powerful to better understand the basic principles for tissue patterning, encompassing the design of synthetic genetic programs encoding cell-cell communication rules and testing their effect on the formation of tissue patterns (Davies, 2017; Li and Elowitz, 2019; Sekine et al, 2018; Toda et al, 2018). This emerging approach is referred to as synthetic developmental biology (Ebrahimkhani and Ebisuya, 2019; Martinez-Ara et al, 2022; Santorelli et al, 2019; Toda et al, 2019). Synthetic morphogen systems have already been developed to explore minimal genetic programs that can encode tissue patterning (Stapornwongkul et al, 2020; Toda et al, 2020). By engineering cells to respond to an inert protein, such as green fluorescence protein (GFP), we and other groups have successfully turned GFP into an effective morphogen capable of inducing target gene expression or activating specific signaling pathways. These studies demonstrated that the trapping of diffusing molecules and activation of intracellular signaling are sufficient for morphogen gradient formation both in vitro and in vivo (Stapornwongkul and Vincent, 2021). However, achieving tissue patterns ultimately relies on the mechanistic cooperation between morphogen diffusion and responsive cell behaviors, such as proliferation and morphological changes (Gilmour et al, 2017). Despite reconstituting the regulatory mechanisms of morphogen gradients (Li et al, 2018; Zhu et al, 2023), there is currently no system to explore the entire patterning process from morphogen diffusion to changes in cell behavior.

To address this, we developed a SYnthetic Morphogen system for Pattern Logic Exploration using 3D spheroids (SYMPLE3D) (Fig. 1A). This system allowed us to study how cellular responses to concentration gradients of secreted molecules could generate tissue domains with sharp boundaries. Here, we found that a circuit in which morphogens induce E-cadherin-based cell adhesion is sufficient to convert a morphogen gradient into a pattern with distinct tissue domains. Despite its simplicity, morphogen-induced E-cadherin serves multiple functions in organizing tissue patterns, including repairing the patterns through cell sorting, generating a uniformly activated domain with tunability, and forming sharp boundaries between tissue domains. Systematic analysis by fine-tuning the GFP concentration revealed a switch-like transition of E-cadherin functions in tissue compaction and cell mixing depending on E-cadherin expression levels, supporting the domain formation process in concert with the morphogen gradient. These findings suggest a potential link between morphogen signaling and differential cell adhesion during pattern formation. In addition, the combination of synthetic morphogens with cell adhesion control provides a new method to design de novo tissue domains for organoid engineering.

## Results

### Constructing synthetic morphogen system in 3D spheroids

To investigate the mechanisms underlying morphogen-based tissue patterning, we constructed a synthetic morphogen system using 3D spheroids (Fig. 1B). We engineered the mouse fibroblast cell line, L929, to prepare the following cells:

(1) GFP secretors: Cells that secrete GFP and express P-cadherin, forming GFP-secreting organizer spheroids.

(2) GFP receiver inducing mCherry (imC cell): Cells that express GFP-anchor protein and anti-GFP synthetic Notch (synNotch) receptor, which recognizes GFP and induce the mCherry reporter (Fig. EV1A).

The GFP-anchor protein is a fusion protein of the GFP nanobody LaG2 and the transmembrane domain (Toda et al, 2020). The synNotch receptor is a Notch-based artificial receptor composed of three components: an extracellular recognition domain, a Notch core transmembrane domain, and an intracellular artificial transcription factor (Morsut et al, 2016). Anti-GFP synNotch has a GFP nanobody LaG17 in the extracellular domain to recognize GFP and TetR-VP64 in the intracellular domain to induce target gene expression.

We separately plated GFP-secretor cells and imC cells in ultra-low-attachment wells to form spheroids. The two types of spheroids were then co-cultured to test how secreted GFP diffused into the spheroid of imC cells (imC spheroid) and generated multicellular patterns (Fig. EV1B). The imC cells captured the secreted GFP, resulting in a consistent GFP gradient that induced a corresponding gradient in mCherry reporter expression (Figs. 1C,D and EV1C, Movie EV1 left). However, only 30% of the co-cultured spheroids maintained an organizer-receiver spheroid topology required to generate a gradient across the imC spheroid. Several imC spheroids engulfed GFP-secretor spheroid during the 48-h culture period (Fig. 2A, Movie EV1, right). The imC spheroid is a loose cell aggregate based on the weak endogenous affinity between L929 cells. L929 cells do not endogenously express classic cadherins such as E-, N-, or P-cadherin, but they do express Pcdh1 and Pcdhgc3, which are protocadherins (Pcdhs) of the cadherin superfamily (Data ref: Sun and Chen, 2020). The default weak cell-cell adhesion in L929 cells could be mediated by these Pcdhs, but this default adhesion is qualitatively different from classic cadherin-based adhesion because Pcdhs lack the binding sites to catenins and cytoskeleton at their cytoplasmic domains (Chen and Maniatis, 2013). To enhance the stability of the organizer-receiver topology, we introduced constitutive expression of E-cadherin in imC cells (imC^Ecad cells) (Fig. 2B). The secretor cells expressing P-cadherin and imC^Ecad cells were clearly segregated by heterophilic adhesion (Bao et al, 2022; Glykofrydis et al, 2021). This modification successfully maintained the organizer-receiver topology for 48 h in all replicates, allowing the formation of synthetic gradients of mCherry induction (Fig. 2C,D, Movie EV2, left). Since the gradient range is determined by the balance between GFP production by secretor cells and GFP consumption by receiver cells (Toda et al, 2020), the gradient range in the compact imC^Ecad spheroid became shorter than that in the loose imC spheroid.

To generate a tissue pattern with a sharp boundary in response to GFP, cells must convert a gradient pattern into distinct tissue domains. We observed that the synthetic gradient contained ectopically activated cells (EACs), which were abnormally activated compared to their surrounding cells (Appendix Fig. S1A,B, indicated by yellow arrowheads). Thus, cells also need to repair these EACs. Thanks to the flexibility of target gene induction in our synthetic morphogen system, we can design and test transcriptional responses to GFP, which enables the organization of tissue domains. Therefore, we named our system SYMPLE3D for general use in studying the design principles of morphogen-based tissue

## A

SYMPLE3D: SYnthetic Morphogen system for Pattern Logic Exploration using 3D spheroids

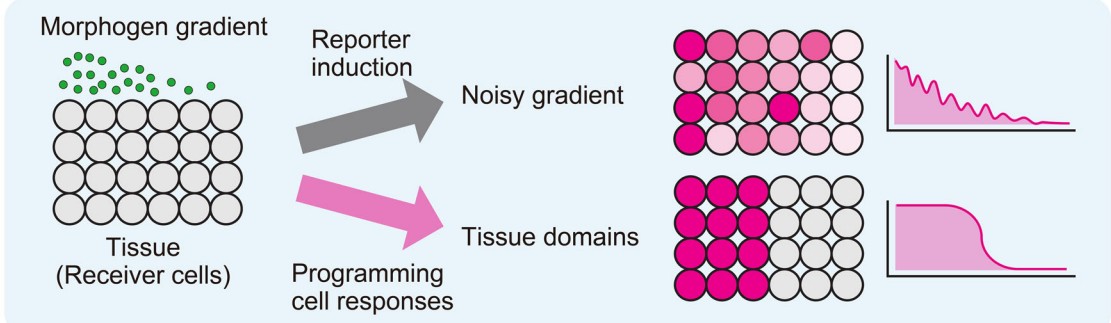

## B

Reconstruct a morphogen system in spheroids

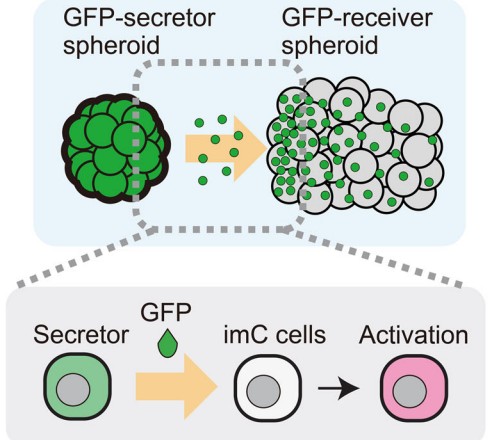

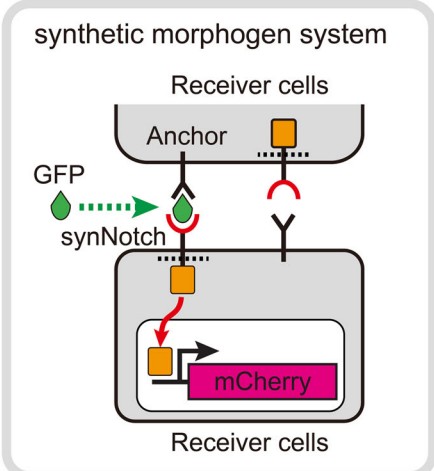

## C

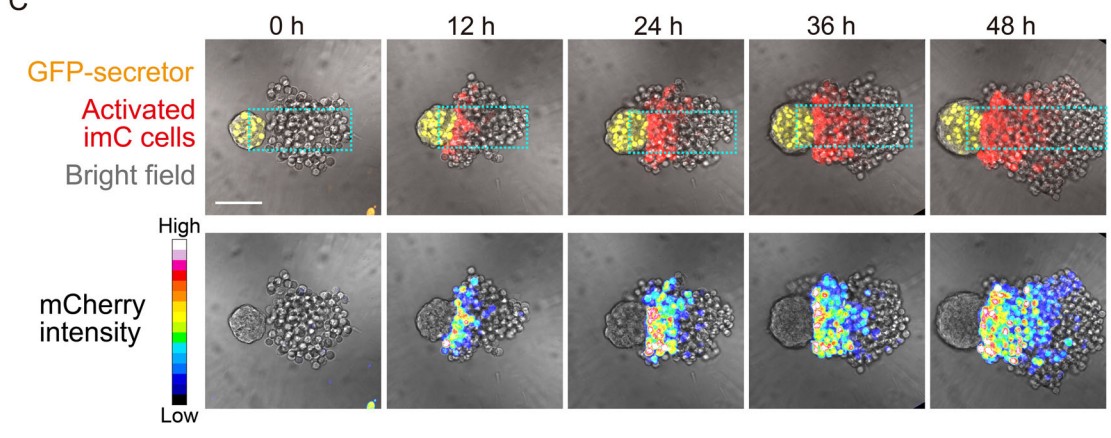

## D

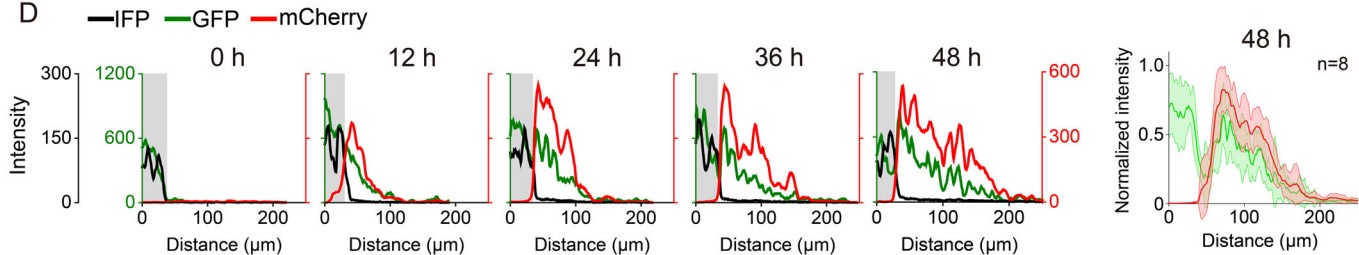

**Figure 1. Synthetic morphogen system for pattern logic exploration using 3D spheroids.**

(A) Concept of SYMPLE3D. Cells sense a morphogen gradient and express a fluorescent reporter, generating a gradient pattern with biological noise. Cell-responsive behaviors to morphogens were programmed to explore the cellular responses sufficient to interpret the morphogen gradient into a tissue pattern with distinct domains. (B) Construction of a morphogen system in 3D spheroids with GFP as a synthetic morphogen. GFP-secretor and GFP-receiver spheroids were co-cultured to observe the diffusion of secreted GFP into the GFP-receiver spheroid, generating multicellular patterns. The GFP secretor spheroid expresses P-cadherin to maintain the topology of the organizer and receiver spheroids. The right panel depicts details of the synthetic morphogen system for imC cells (Toda et al, 2020). GFP is trapped on the cell surface with a GFP-anchor protein and recognized by the anti-GFP synNotch to induce the mCherry reporter. (C) Time course of synthetic gradient formation in imC spheroids. GFP-secretor spheroids with 50 GFP-secretor cells and GFP-receiver spheroids with 200 imC cells were separately formed and co-cultured at time 0 h. Spheroid images acquired at 0–48 h using a confocal microscope. GFP secretor cells express nucleus-localized IFP2.0, shown as a yellow pseudocolor. Scale bar: 100 μm. mCherry distributions are visualized by 16 pseudocolors in the bottom images. Further details are provided in Movie EV1. (D) Fluorescence intensity profiles of imC cells within rectangular regions (green line: GFP, red line: mCherry, black line: IFP, indicating the region of GFP secretor cells). Multiple profiles of snapshots taken at 48 h with a 20x WI objective are overlaid in the rightmost graph. The means are indicated with bold lines (green line: GFP, red line: mCherry). Shaded areas in the graph represent ± standard deviation (SD). $n = 8$. Source data are available online for this figure.

patterning. Prior to this, we estimated the GFP concentration in the spheroids to ensure that GFP production is not excessive to cause signal saturation and alter tissue patterning. We confirmed that the GFP concentration around the GFP-secretor spheroids was approximately 2 nM, lower than the saturation concentration of the receiver cells (Appendix Fig. S2A,B).

## Tissue domain formation by coupling morphogen signal with cell adhesion

To address the issue of EACs in the synthetic gradient pattern, we hypothesized that inducing cell adhesion could spatially sort EACs into positions where surrounding cells express similar levels of cell adhesiveness. Based on this concept, we engineered GFP-receiver cells to induce mCherry-fused E-cadherin (E-cadherin-mCherry), termed iEcad-mC cells and co-cultured iEcad-mC spheroids with GFP-secretor spheroids (Fig. 3A; Appendix Fig. S3A). However, beyond our hypothesis, two distinct domains with a sharp boundary emerged instead of a gradient pattern (Fig. 3B,C, Movie EV2, right). Unlike the mCherry gradient patterns observed in imC and imC^Ecad cells (Figs. 1D and 2D), the induced E-cadherin-mCherry remained high from the interface with the GFP-secretor spheroid to the middle of the iEcad-mC spheroid and then sharply decreased, resulting in the segmentation of iEcad-mC spheroids into mCherry-positive and -negative domains with a sharp boundary. Confocal microscopy of 3D spheroid structures confirmed the compartmentalization of the E-cadherin-mCherry-positive domain in the iEcad-mC spheroids (Fig. 3D). We quantified the steepness of the mCherry intensity slope in imC, imC^Ecad, and iEcad-mC spheroids using the Hill coefficient, revealing that the boundary between the mCherry-positive and -negative regions in iEcad-mC spheroids was significantly sharper than the mCherry gradients in imC and imC^Ecad spheroids (Fig. 3E; Appendix Fig. S3C). These findings demonstrate that coupling morphogen signaling with cell adhesion is sufficient to interpret the GFP gradient into distinct active and inactive domains in our SYMPLE3D system.

Furthermore, we examined the robustness of this pattern formation process against variable culture conditions. First, to investigate how cell division could affect the spreading of GFP and activated region during the pattern formation, the spheroids were incubated in a low serum concentration media (0.5% FBS), which is much lower than normal culture condition (10% FBS) to reduce the cell division rate. As a result, the division rate of iEcad-mC

cells was significantly reduced to form a smaller spheroid after 48 h culture in the low serum media, but they formed a similar sharp domain even in that condition (Fig. EV2A,B). While the reduction of serum concentration could alter not only cell division but also other cell properties such as metabolism, the pattern formation process based on the combination of morphogen and cell adhesion is robust against such changes. The boundary between active and inactive domains in low serum conditions may appear sharper than in normal conditions, but further studies are required to clarify how the cell division rate affects the boundary sharpness. In addition, in tissue engineering applications, it is essential to robustly form tissue domains of desired sizes. We explored the tunability of synthetic tissue domain sizes using various numbers of GFP-secreting cells (Appendix Fig. S4A). As the initial number of GFP-secreting cells increased from 25 to 200, the size of the activated domain in the iEcad-mC spheroids varied in proportion to the number of GFP-secreting cells (Appendix Fig. S4B). We quantified the domain size for each condition using the $k$ value of the Hill equation (Appendix Fig. S4C), demonstrating that the morphogen-induced adhesion circuit allows for the precise formation of synthetic tissue domains with tunable sizes by adjusting the amount of morphogen input.

## Continuous cell sorting and switch-like tissue compaction generate tissue domains

To gain insight on how the diffusible GFP signal forms tissue domains by inducing E-cadherin, we analyzed the time course of domain formation with high temporal resolution (Movie EV3). At approximately 6 h, initially activated iEcad-mC cells scattered within the spheroids began to aggregate into a single domain due to E-cadherin induction (Fig. 4A). Furthermore, EACs were generated at distant positions from the GFP-secretor spheroid but were eventually sorted and absorbed into the E-cadherin-positive domain (Fig. 4B; Appendix Fig. S3D, Movie EV3). This suggests that E-cadherin-inducing cells can maintain a robust single domain with the sorting-based pattern correction, even in the presence of EACs continuously emerging.

The E-cadherin-mCherry-positive domain was easily distinguishable from the inactive domain, even in the bright-field images, as cell outlines became blurred owing to E-cadherin-mediated cell compaction (Appendix Fig. S3B). Although differential cadherins are known to generate sharp tissue boundaries (Halbleib and Nelson, 2006), it remains unclear why the gradient signal of

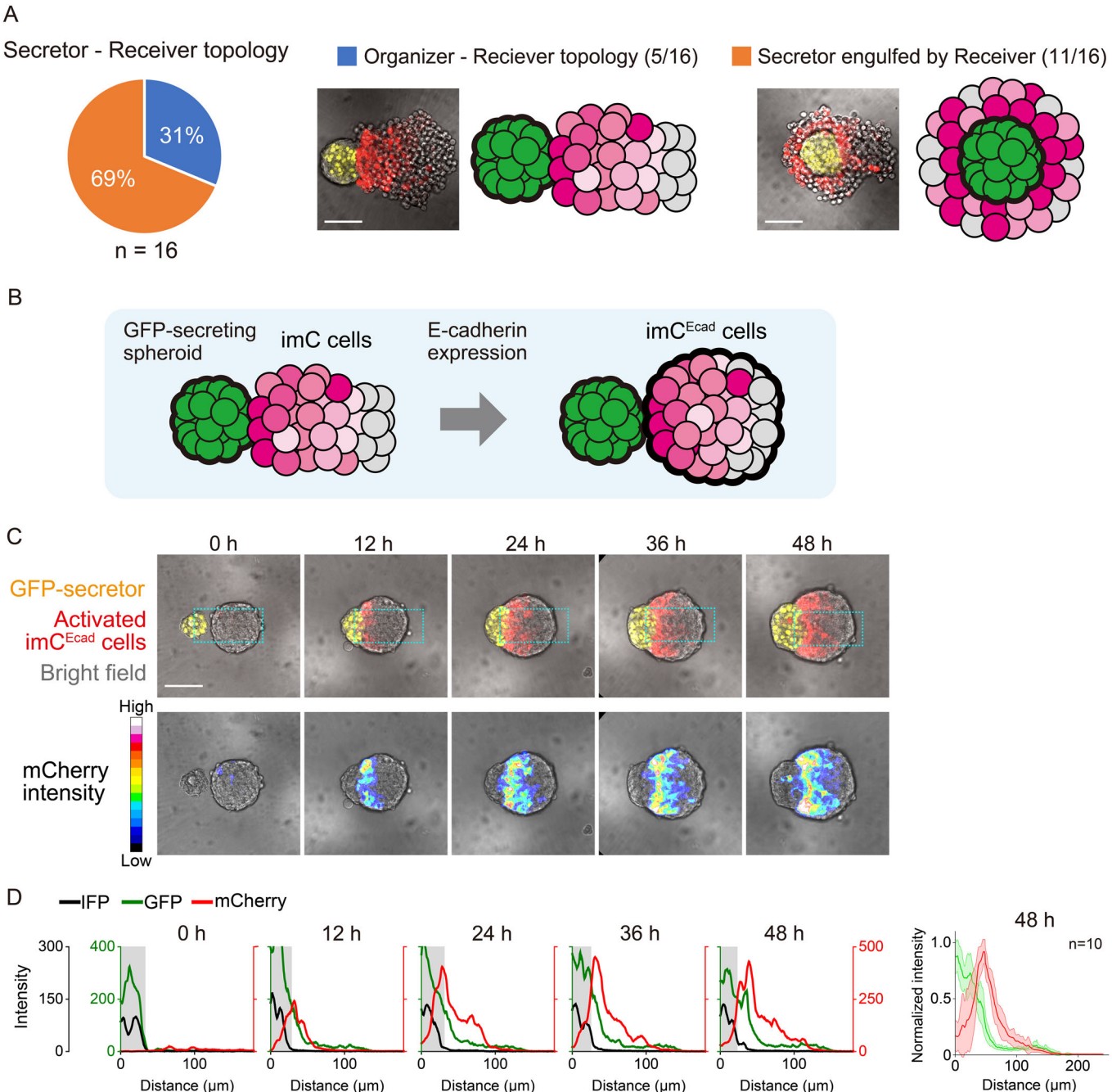

**Figure 2. E-cadherin expression maintains the organizer-receiver topology for synthetic gradient formation.**

(A) Ratio of organizer-receiver topology in the co-culture system of imC and GFP-secretor spheroids. Approximately 30% (5/16) of the imC spheroids maintained the organizer-receiver topology, whereas the remaining 70% (11/16) engulfed the GFP-secretor spheroid and lost the organizer-receiver topology. Scale bar: 100 μm. (B) E-cadherin expression was used to form a compact spheroid of imC cells, which ensured the maintenance of the organizer-receiver topology for 48 h. (C) Time course of synthetic gradient formation in imC$^{Ecad}$ spheroids. The imC$^{Ecad}$ spheroid maintained its organizer-receiver topology for 48 h of co-culture with a GFP-secretor spheroid. A synthetic gradient gradually developed in the compact imC$^{Ecad}$ spheroid. Scale bar: 100 μm. mCherry distributions are visualized by 16 pseudocolors in the bottom images. Further details are provided in Movie EV2. (D) Fluorescence intensity profiles of imC$^{Ecad}$ cells within rectangular regions (green line: GFP; red line: mCherry; black line: IFP indicating the region of GFP-secretor cells). Multiple profiles of snapshots taken at 48 h with a 20x WI objective are overlaid in the rightmost graph. The means are indicated with bold lines (green line: GFP, red line: mCherry). Shaded areas in the graph represent ± SD. $n = 10$. Source data are available online for this figure.

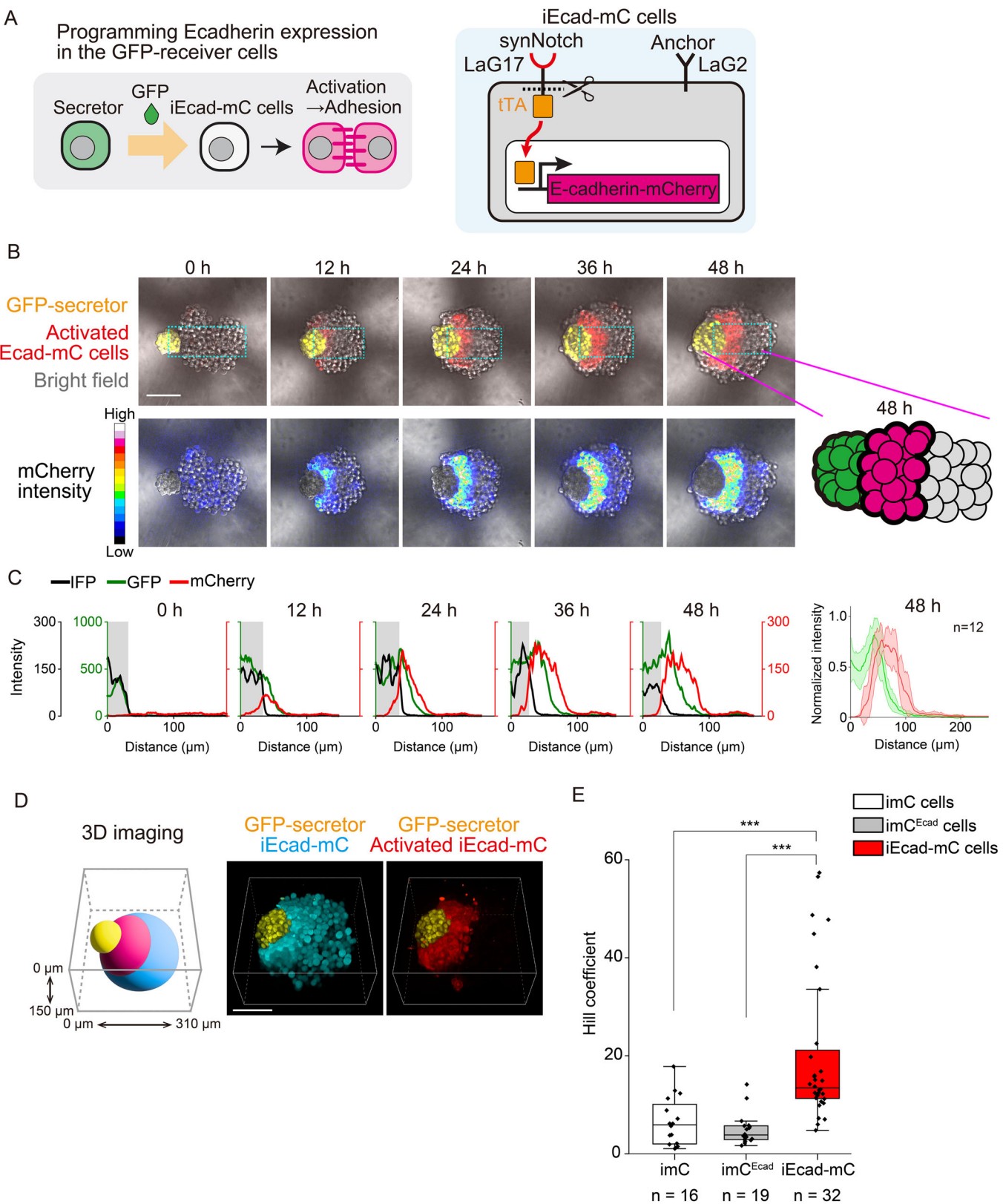

Figure 3. **Tissue domain formation by coupling morphogen signaling with cell adhesion.**

(A) Induction of mCherry-fused E-cadherin (E-cadherin-mCherry) in iEcad-mC cells in response to GFP. (B) Time course of tissue domain formation via morphogen-induced adhesion. GFP-secretor spheroids with 50 GFP-secretor cells and GFP-receiver spheroids with 200 iEcad-mC cells were co-cultured for 48 h. Activated iEcad-mC cells accumulate next to GFP-secretor spheroids, and the iEcad-mC spheroids eventually segmented into distinct active and inactive domains. Scale bar: 100 μm. mCherry distributions are visualized by 16 pseudocolors in the bottom images. Further details are provided in Movie EV2. (C) Fluorescence intensity profiles of iEcad-mC cells within rectangular regions (green line: GFP; red line: mCherry; black line: IFP, indicating the region of GFP-secretor cells). Multiple profiles of snapshots taken at 48 h with a 20x WI objective are overlaid in the rightmost graph. The means are indicated with bold lines (green line: GFP; red line: mCherry). Shaded areas in the graph represent ± SD. $n = 12$. (D) 3D observation of the synthetic tissue domain. The co-cultured GFP-secretor and iEcad-mC spheroids imaged at 48 h using confocal microscopy to reconstruct 3D spheroid structures. Scale bar: 100 μm. The middle panel shows BFP-expressing iEcad-mC cells with cyan pseudocolor. The right panel shows activated iEcad-mC cells expressing E-cadherin-mCherry with a red pseudocolor, forming an activated domain with a sharp boundary from the inactive cell population. (E) Quantification of boundary sharpness between mCherry-positive and mCherry-negative regions. The spatial mCherry profiles inside the rectangular regions were fitted to the Hill equation using OriginPro software (Appendix Fig. S3C). The Hill coefficients are plotted with their mean ± SD. The boxes represent the group median and interquartile range (25th–75th percentiles). The whiskers extend to the minimum and maximum data points within 1.5 times the interquartile range from the 25th and 75th percentiles, while data points beyond this range are considered outliers. Differences between two groups were determined using Welch's t-test with *** for $P < 0.001$. ($P = 4.4 \times 10^{-5}$ between imC and iEcad-mC cells. $P = 4.4 \times 10^{-6}$ between imC$^{Ecad}$ and iEcad-mC cells.) imC: $n = 16$, imC$^{Ecad}$: $n = 19$, iEcad-mC: $n = 32$. Source data are available online for this figure.

E-cadherin induction generates a sharply compact domain. To address this question, we used purified GFP to systematically analyze the compaction of iEcad-mC cells induced by variable GFP concentrations, mimicking various GFP levels within the GFP gradient. Increasing the GFP concentration from 0 to 2 nM promoted proportional increases in E-cadherin and mCherry induction levels (Figs. 4C and EV3A–D). However, in iEcad-mC cells, the spheroid compaction was fully induced at approximately 0.1 nM GFP, and no further compaction occurred at higher GFP concentrations (Fig. 4C). This suggests that E-cadherin induces tissue compaction in a switch-like manner in response to E-cadherin expression levels. Since the synthetic GFP gradient started at approximately 2 nM (Appendix Fig. S2B), most activated cells could sense more than 0.1 nM GFP, which is enough to trigger compaction, resulting in the formation of a single compact domain. In summary, morphogen-induced E-cadherin forms a distinct tissue domain with cell sorting and switch-like compaction (Fig. 4D).

## Mixing of E-cadherin-inducing cells inside the GFP gradient leads to the formation of a uniformly activated tissue domain

Another intriguing aspect of the synthetic tissue domain was the uniformity of activation levels. As shown in Fig. 3B, the distribution of induced E-cadherin-mCherry in the activated domain was uniformly high, whereas GFP distribution showed a gradient. The estimated GFP concentration gradient ranged from 2 to 0 nM (Appendix Fig. S2B), whereas iEcad-mC cells induced various levels of E-cadherin-mCherry expression in response to the same concentration range of GFP (Fig. 4C). Thus, the reason behind the sustained high induction level of E-cadherin–mCherry in this domain remains unclear. To investigate how this uniformity arises, we examined how iEcad-mC cells behave in the presence of cells expressing high levels of E-cadherin and parental L929 cells (Ecad-high cells and no-Ecad cells, respectively). We mimicked the conditions of iEcad-mC spheroids inducing variable levels of E-cadherin within the GFP gradient by adding the variable concentrations of purified GFP. In this assay, Ecad-high cells formed a compact spheroid surrounded by no-Ecad cells, and the positions of iEcad-mC cells were dependent on the amount of GFP. With a small amount of GFP, activated iEcad-mC cells were

attached to the surface of Ecad-high compact spheroids. However, in the presence of more GFP (greater than 0.1 nM), the activated iEcad-mC cells mixed into the Ecad-high compact spheroids (Fig. EV4A–C). We also observed the time course of cell mixing with a stimulus of 0.0625 nM GFP, which was close to the threshold level (Fig. 5A, Movie EV4). iEcad-mC cells were gradually activated and attached to the surface of Ecad-high spheroids at an early time point. Some iEcad-mC cells then started to penetrate the compact spheroid. To estimate how much difference in E-cadherin expression levels between Ecad-high cells and iEcad-mC can be tolerated to induce cell mixing, we measured their E-cadherin levels with immunostaining (Fig. EV4D). The data showed that the E-cadherin induction levels in iEcad-mC cells increased proportionally with the increase of GFP concentration. At the concentration 0.0625 nM GFP, where iEcad-mC cells began to mix with Ecad-high cells, there was approximately a 35-fold difference between Ecad-high and iEcad-mC cells on the immunostaining-based flow cytometry analysis. These results indicate that approximately a 35-fold difference between two cell types can be tolerated to induce the cell mixing behavior. When the fold-difference was higher with the lower concentration of GFP (0.03125 nM GFP), most iEcad-mC cells attached to the surface of the Ecad-high cells' spheroid and exhibited less mixing (Fig. EV4B), suggesting that the E-cadherin expression range should be less than 35-fold to induce cell mixing.

We also found that E-cadherin-expressing cells could move fluidly in their compact spheroids by tracking the fluorescence-labeled cells immediately after cell division in the E-cadherin-positive spheroid (Fig. 5B, Movie EV5). Interestingly, when we observed the pattern formation in a 2D culture system to reduce the cell motility by attaching cells on the tissue culture-treated plate, iEcad-mC cells formed a gradient pattern similar to that of imC cells and did not induce a distinct domain with a sharp boundary (Fig. EV5A,B). This result suggested that cell motility and frequent cell rearrangement are essential for creating the observed pattern. These findings indicate that E-cadherin enables cell mixing instead of statically fixing cell positions within the activated domain. Taken together, when iEcad-mC cells sense more than the threshold level of GFP in the GFP gradient, iEcad-mC cells can penetrate the region where cells express higher levels of E-cadherin, mixing the activated iEcad-mC cells to form the uniform domain (Fig. 5C).

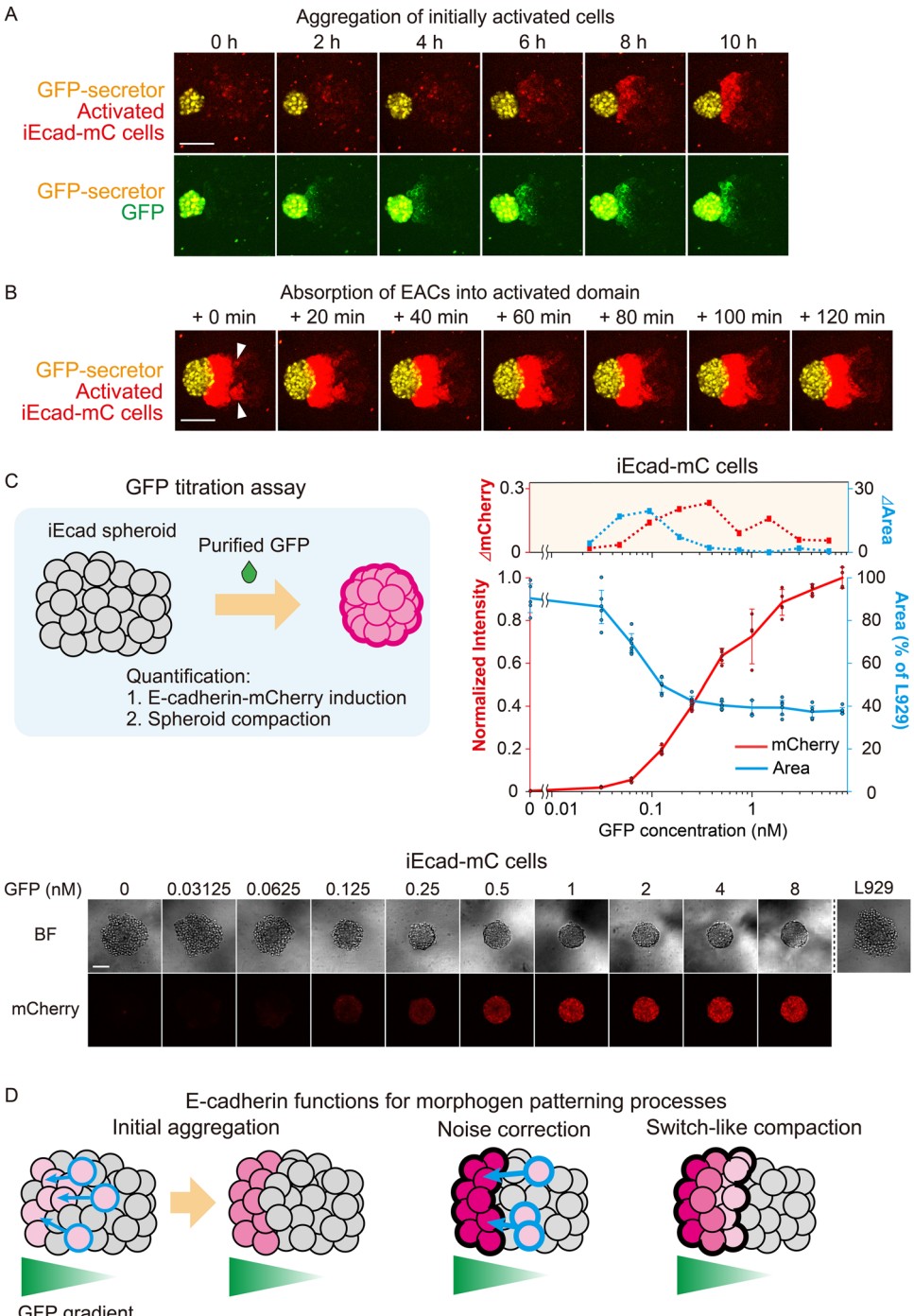

**Figure 4. Continuous cell sorting and switch-like compaction lead to domain formation.**

(A) Aggregation of the initially activated iEcad-mC cells. At early time points, spreading GFP is captured on iEcad-mC cells, inducing E-cadherin-mCherry expression and aggregation of the initially activated iEcad-mC cells. Further details are provided in Movie EV3. Scale bar: 100 μm. $n = 12$. (B) Absorption of EACs into a single activated domain. Images showing enhanced mCherry signal, allowing visualization of the EACs at the edge of the GFP gradient. Some EACs were generated separately from the activated domain (arrowheads). However, such EACs were sorted and absorbed into already existing activated domains, resulting in the formation of a single activated domain. Further details are provided in Movie EV3. Scale bar: 100 μm. $n = 12$. (C) Switch-like compaction of iEcad-mC spheroids in response to low GFP concentrations. iEcad-mC spheroids containing 100 iEcad-mC cells cultured with various concentrations of purified GFP. In the right graph, the left and right numbers indicate the normalized fluorescence intensity of E-cadherin-mCherry and the percentage of spheroid area relative to L929 spheroids, respectively. The difference in the values between adjacent GFP concentrations is plotted in the top graph. Bright-field and mCherry images of the iEcad-mC spheroids are shown at the bottom. Experiments were performed with 4–6 replicates, and data are presented as mean ± SD. (D) Schematic representation of the potential role of induced E-cadherin expression in morphogenic tissue patterning. Source data are available online for this figure.

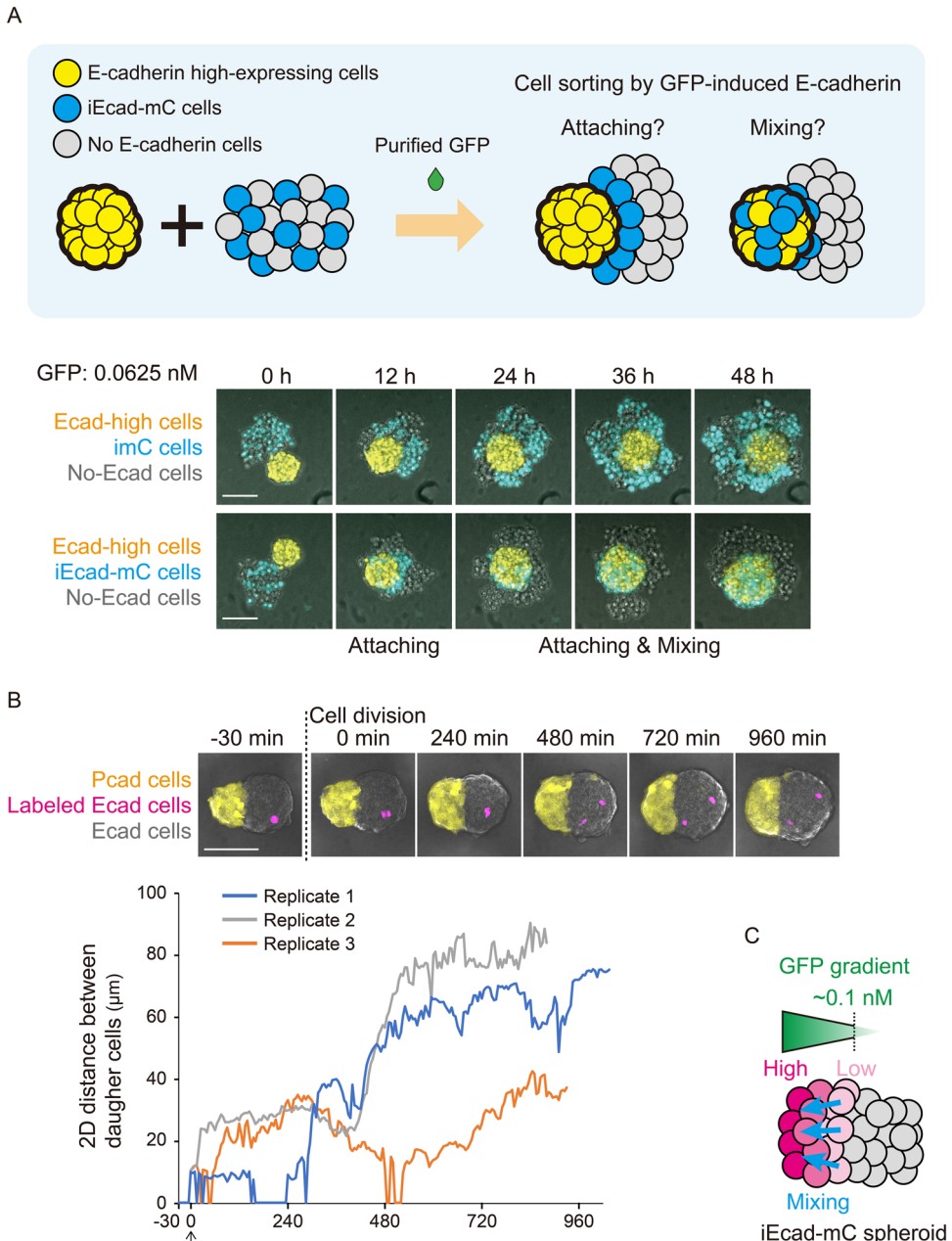

**Figure 5. E-cadherin-induced cell mixing and dynamic behaviors in compact spheroids.**

(A) Mixing of high-E-cadherin-expressing cells (Ecad-high cells) and activated iEcad-mC cells above the threshold level. Spheroids with 100 E-cadherin high-expressing cells mixed with spheroids made by a mixture of 30 imC or iEcad-mC cells and 30 parental L929 cells (no-Ecad cells) in the presence of 0.0625 nM GFP (the approximate threshold level for spheroid compaction). E-cadherin high cells are labeled in yellow, and imC or iEcad-mC cells are labeled in cyan. Temporal dynamics of imC or iEcad-mC cell distribution in spheroids are shown. Further details are provided in Movie EV4. Scale bar: 100 μm. $n = 6$. (B) E-cadherin-expressing cells move randomly in compact spheroids. E-cadherin-expressing cells (Ecad cells) and E-cadherin-expressing cells with nucleus-localized IFP2.0 (labeled Ecad cells) were mixed at a ratio of 199:1 to form a spheroid. This spheroid was then co-cultured with 50 P-cadherin-expressing cells (Pcad cells) as a reference to monitor spheroid movement. Yellow cells, pink cells, and non-color cells are Pcad cells, labeled Ecad cells, and Ecad cells, respectively. Spheroid dynamics imaged every 5 min for 24 h. Scale bar: 100 μm. At time 0 min, IFP2.0-expressing cells undergo cell division, after which the movement of each daughter cell in the maximum intensity projection images is tracked, and the distance between cells measured. The blue line indicates the distance between the daughter cells shown in the top images. This experiment was conducted using three replicates of this trace experiment (blue, gray, and orange tracings). Further details are provided in Movie EV5. (C) Potential cell mixing in the iEcad-mC spheroid in Fig. 3. Weakly-activated cells that express low-level of E-cadherin could penetrate into the highly-activated region where cells express higher-level of E-cadherin. Source data are available online for this figure.

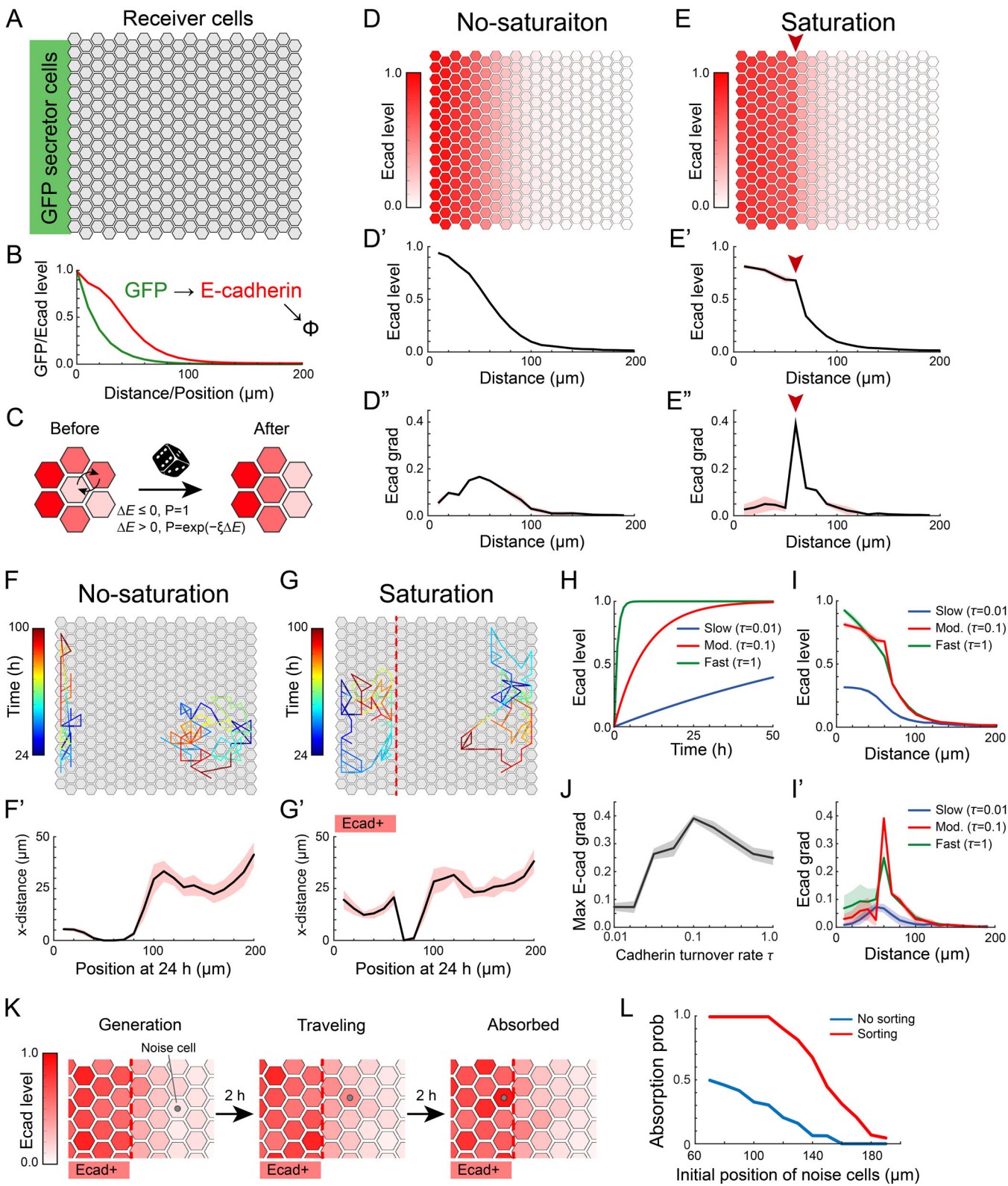

Figure 6.   Mathematical model of tissue domain formation with morphogen-induced cell adhesion.

(A) Schematic representation of the simulation domain. GFP source cells (green) are located on the left, and hexagonal receiver cells are arranged on the right. (B) Spatial distribution of GFP level (green) and E-cadherin expression level (red) along the x-axis of the domain in the case of no cell movement at the steady state. (C) Schematics depicting the rule of cell swapping. Cell swapping occurs deterministically when the energy difference between before and after the swapping is non-positive ($\Delta E \leq 0$), while it occurs stochastically when the energy difference is positive ($\Delta E > 0$). (D, E) Spatial distribution and gradient of E-cadherin expression level at 48 h in the no-saturation regime (D–D") and in the saturation regime (E–E"). Arrowheads indicate the position where the gradient is maximized, corresponding to the sharp boundary (E', E"). The means are indicated with bold lines. Shaded areas in the graphs represent ± SD. $n = 10$. (F, G) Tracing of representative cells and their traveling distance along x-axis in the no-saturation regime (F–F') and in the saturation regime (G–G'). The tracer colors represent the time. Ecad+ represents the E-cadherin-positive domain. The x-distance was assessed by averaging positions between 24 h and 100 h over y-axis. The means are indicated with bold lines. Shaded areas in the graphs represent ± SD. $n = 10$. (H) E-cadherin expression level over time with the different values of the turnover parameter τ. (I, I') E-cadherin expression (I) and its gradient (I') with the different rates τ at 48 h. The means are indicated with bold lines. Shaded areas in the graphs represent ± SD. $n = 10$. (J) Maximum values of E-cadherin gradient over different turnover rates τ. The mean is indicated with a bold line. Shaded areas in the graphs represent ± SD. $n = 10$. (K) Snapshots showing the process of noise cell generation to the domain absorption. (L) Absorption probability over the position of noise cells generated. Average population number at each position is 39. Source data are available online for this figure.

## Mathematical modeling of tissue domain formation by morphogen-induced adhesion

Finally, we tested our experimental findings with a simple mathematical model. The model represents the essential spatial arrangement of cells in our experimental system, where GFP source cells and receiver cells are segregated in a two-dimensional space (Fig. 6A). Secreted GFP, diffusing through the tissue, induces E-cadherin expression in the receiver cells, as determined from our experimental measurements (Appendix Fig. S5A), and E-cadherin is subject to decay with the turnover parameter τ. These conditions contribute to establishing a stable distribution within the tissue in the absence of cell movement (Fig. 6B). In line with our experimental observations, E-cadherin-expressing cells are spatially arranged through the cell sorting based on differential E-cadherin expression (Fig. 4D). Moreover, the receiver cells exhibit frequent positional changes over time within their cluster (Fig. 5B, Movie EV5), suggesting that cell movement is primarily governed by differential adhesion energy (DAE), defined as the difference in E-cadherin expression levels between neighboring cells, along with random cell movement. Accordingly, we modeled cell swapping, guided by E-cadherin-based DAE $\Delta E$ between neighboring cells, with stochasticity, thereby gradually facilitating the transition towards the state of global minimum energy (Fig. 6C). As shown in Fig. EV4B–D, the effective E-cadherin-based cell sorting is abolished when the E-cadherin level exceeds the threshold level; we implemented this saturation regime of DAE into the $\Delta E$ function in our model.

We first examined whether the sharp boundary of the E-cadherin activated domain could be reproduced through simulations of the model. To emphasize the impact of DAE saturation regime on the E-cadherin pattern, we contrasted it with a no-saturation regime, where DAE depends on the differential E-cadherin level without saturation. In the no-saturation regime, the E-cadherin profile exhibits a smooth gradient over space (Fig. 6D–D"), similar to the gradient profile without cell motion (Fig. 6B). However, in the saturation regime, the E-cadherin profile shows a well-defined domain with a steep gradient at its edge (Fig. 6E–E"; Appendix Fig. S5B). The peak position of the E-cadherin gradient corresponds to the position with the GFP threshold in the saturation regime. In addition, the model can quantitatively reproduce the experimental data regarding tunability of E-cadherin domain size (Appendix Fig. S5C). These results demonstrate that the DAE saturation regime gives rise to a tunable E-cadherin-positive domain with a distinct sharp boundary.

How does the saturation regime of DAE generate the sharp boundary? We hypothesized that the devoid of DAE in the saturation regime leads to mixing behavior of cells, resulting in a reduced E-cadherin spatial gradient within the E-cadherin-positive domain. To investigate this, we tracked the cells over time and observed that cells within the E-cadherin-positive domain exhibited greater travel distances along the x-axis in the saturation regime compared to those in the non-saturated regime (Fig. 6F–G'). No cell movement along the x-axis was observed at the position one-cell away from the threshold position due to the effective E-cadherin-based DAE (Fig. 6G'), indicating that the cell mixing is confined within the E-cadherin-positive domain.

Given the critical role of cell mixing, the turnover of E-cadherin expression is expected to influence the sharp boundary formation. A faster turnover relative to the cell mixing timescale could lead to E-cadherin distribution that more closely reflects the GFP spatial profile. In our simulations, we set the standard parameter value $\tau = 0.1$ as E-cadherin reaches a steady state within 1–2 days (Figs. 3B,C and 6H). To explore the impact of turnover rate, we investigated two additional scenarios: a fast turnover rate ($\tau = 1$) and a slow turnover rate ($\tau = 0.01$). In the case of fast turnover, the E-cadherin expression exhibits a smooth gradient profile (Fig. 6I,I'), similar to that observed without cell movement (Fig. 6B). In contrast, in the case of slow turnover, the E-cadherin expression level remains lower compared to the other scenarios (Fig. 6I) since it does not reach a steady state even at 48 h (Fig. 6H), consequently resulting in a lower gradient (Fig. 6I'). Remarkably, the intermediate turnover rate demonstrates the maximum E-cadherin gradient (Fig. 6J), indicating that a moderate turnover rate of E-cadherin as observed in the experiments achieves the sharpest gradient profile of the E-cadherin-positive domain.

Lastly, we examined whether the model could effectively capture the pattern correction through sorting. We assumed stochastic generation of EACs, which received the GFP input with higher sensitivity in the E-cadherin-negative region. Notably, in our simulations, these EACs gradually approached and eventually became absorbed into the E-cadherin-positive domain (Fig. 6K), mirroring the observations from our experiments (Fig. 4B, Movie EV3). The probability of absorption was found to be higher for EACs that were closer in birth position to the E-cadherin-positive region (Fig. 6L). Moreover, regardless of the birth position, the absorption probability significantly exceeded that of a control situation without cell sorting (Fig. 6L; Appendix Fig. S5D, Movie EV6). The collective evidence

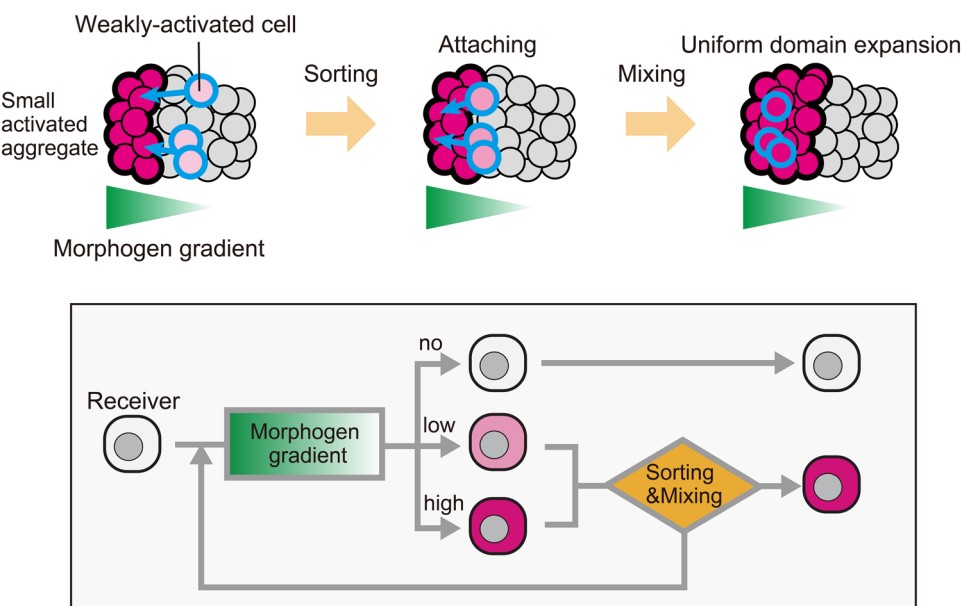

**Figure 7. Working model of tissue domain formation with morphogen-induced cell adhesion.**

This schematic representation illustrates the process of tissue-domain formation mediated by morphogen-induced adhesion. Cells located close to the GFP-secretor spheroids are strongly activated (red), leading to the formation of aggregates. On the other hand, cells at the edge of the GFP gradient receive a lower GFP concentration, resulting in weaker activation (pink). These weakly activated cells induce low levels of E-cadherin, triggering their sorting and integration into strongly activated aggregates. As the weakly activated cells express E-cadherin above a threshold level, they begin to mix with the strongly activated domain. Consequently, these cells penetrate the domain and move towards the GFP-secretor spheroid, where they are exposed to higher GFP concentrations. Upon entering the highly activated aggregate, the cells become further activated, ultimately leading to the formation of a uniformly activated domain. The cell lineage dynamics depicted in this model are represented in the bottom scheme.

supports the validation of our model, built upon experimental data, reproducing the robust pattern formation within the tissue.

## Discussion

In this study, we developed the SYMPLE3D system, an in vitro platform using a spheroid culture, to investigate design principles of how morphogens generate tissue patterns in a 3D tissue context. Utilizing an orthogonal diffusible ligand-receptor system with GFP as an artificial morphogen, we explored the minimum mechanisms of morphogen interpretation without activating intrinsic signaling pathways.

Our focus was on the interplay between morphogen gradients and cell adhesion in tissue patterning. While these mechanisms have been studied separately in various model systems, recent studies on neural tube patterning showed that the Shh gradient controls the expression of cadherin families to organize cells into multiple sharp domains (Tsai et al, 2020; Xiong et al, 2013). These studies showed the importance of integrating morphogen gradients with differential cell adhesion for robust pattern formation. However, it is challenging to prove the patterning principles of morphogen-driven cell adhesion when morphogens induce various changes in cellular properties simultaneously in vivo. Here, SYMPLE3D system allowed the proof-of-principle experiments in a simplified multicellular system, where the synthetic morphogen controls only cadherin expression. We showed that the straightforward coupling morphogen signaling with cell adhesion is sufficient to transform a gradient into distinct tissue domains. In our working model, weakly activated receiver cells at the

edge of the morphogen gradient (pink cells) were sorted to attach to cadherin-high domain (red cells) (Fig. 7). When receiver cells express cadherin above a threshold level, they begin to penetrate and mix with the cadherin-high domain. Through cell sorting and mixing, cells approach the GFP-secretor spheroid and receive more GFP, leading to an acceleration of cadherin induction and expansion of the cadherin-high domain. Collectively, the cell sorting and mixing of activated cells within the gradient generate a uniformly activated domain. This result suggests that changes in cell adhesion properties during morphogen-induced differentiation could be involved in tissue-patterning processes. The simplicity of this interplay also suggests that the generation of new tissue domains by coupling diffusing molecules with cell adhesion control might occur through evolution.

Our SYMPLE3D system is extensible to investigate the pattern formation logic in various cases. First, since natural tissues use multiple morphogens to define complex positional information, such as a combination of BMP, Wnt, and Shh (Briscoe and Small, 2015), we can introduce multiple synthetic morphogens into the system. We previously developed another synthetic morphogen based on a fusion protein of mCherry and a PNEtag (Toda et al, 2020), allowing us to create two independent gradients. In addition, we can also program diverse cellular responses downstream of the synNotch receptor to morphogens. For instance, gene regulatory networks are necessary for the precise interpretation of morphogen amounts into different cell fates (Exelby et al, 2021; Meinhardt, 2009). This allows us to explore possible morphogen-based circuits for tissue patterning by not only reconstituting natural complex patterns but also creating new genetic programs for desired tissue

architectures (Elowitz and Lim, 2010; Martinez-Ara et al, 2022). Moreover, we can introduce the SYMPLE3D experimental setup into different cell types such as embryonic stem cells in which the synNotch receptor works (Malaguti et al, 2022; Morsut et al, 2016).

Our synthetic systems also enable the systematic analysis of E-cadherin function by fine-tuning the induction levels of E-cadherin with GFP titration. We observed a switch-like behavior of E-cadherin on tissue compaction and cell mixing. E-cadherin has a threshold-like expression level that induces tissue compaction, which is crucial to generate an E-cadherin-positive domain with a sharp boundary. Moreover, when the two cell types express E-cadherin at different levels, they can mix as long as both express E-cadherin above the threshold level. This mixing process facilitates the penetration of weakly activated E-cadherin-low cells into highly activated E-cadherin-high regions within the iEcad-mC spheroid, resulting in enhanced activation of the weakly activated cells and the formation of a uniformly activated domain. However, the molecular mechanisms underlying the switch-like behavior of E-cadherin require further investigation. While E-cadherins bind to each other via its homophilic extracellular domain, their intracellular domain, which binds to adaptor proteins connected to the actin cytoskeletons, also plays a role in tissue compaction and cell sorting (Maitre et al, 2012). Catenin adaptor proteins have been shown to regulate actin assembly and dynamics at cell-cell contacts (Shapiro and Weis, 2009). In addition to the extracellular affinity, these intracellular signals may enable spheroid compaction and cell mixing with different E-cadherin expression levels when cells express E-cadherin at levels higher than those enough to activate the intracellular signals. Moreover, the mechanosensitive recruitment of the adaptor proteins at the E-cadherin intracellular domain could contribute to this cell mixing (Noordstra et al, 2023; Yonemura et al, 2010). A further study is required to characterize the details of the cytoskeletal and signaling features as well as incorporating additional ingredients, such as cell morphology, and mechano-chemical effects to the mathematical model.

Genetically programming tissue patterns can aid in modifying morphogenetic processes of organoid systems (Hofer and Lutolf, 2021; Trentesaux et al, 2023), forming a new gradient pattern that induces the genes of interest to spatially guide cell differentiation. However, if the synthetic gradient defines a rough pattern with EACs and unclear boundaries, it loses its applicability. Our findings suggest the possibility of programming a new tissue domain with sharp boundaries in organoids by combining synthetic morphogens with cell adhesion control. Recently, it was reported that the knockdown of E-cadherin in a subpopulation of human iPS cells induces cell sorting to form mosaic patches of wild-type and knockdown cells (Libby et al, 2018). In addition, technologies to induce transcriptome regulation downstream of synNotch have also been reported (Lee et al, 2023). By controlling cell adhesion and transcription factors with synthetic morphogens, it may be feasible to design de novo tissue domains with specific cell fate induced in iPS cells. Utilizing synthetic organizer cells or a microfluidic apparatus to administer synthetic morphogens, we can establish a synthetic axis featuring contiguous domains of specific cell fates, which will help spatially organize symmetry breaking and differentiation, currently uncontrolled in conventional organoid cultures. SYMPLE3D provides a new synthetic biology approach for mechanistically studying tissue patterning and engineering organoid structures.

# Methods

## Reagents and tools table

| Reagent/Resource | Reference or Source | Identifier or Catalog Number |
|---|---|---|
| **Experimental models** | | |
| L929 (*M. musculus*) | RIKEN BRC, Cell Bank | RCB1422 |
| K562 (*H. sapiens*) | ATCC | CCL-243 |
| **Recombinant DNA** | | |
| pHR_TRE3GS-->Ecad-mCherry_PGK-->BFP | This study | N/A |
| pHR_EF1a-->myc-LaG17-tTA synNotch | Addgene | 162230 |
| pHR_EF1a-->HA-mtLaG2 | Addgene | 162225 |
| pHR_TRE3GS-->mCherry_PGK-->BFP | Addgene | 162231 |
| pHR_EGFPligand | Addgene | 79129 |
| pHR_EF1a-->Ecad | This study | N/A |
| pHR_EF1a-->sGFP | This study | N/A |
| pHR_EF1a-->NLS-IFP-P2A-Pcad | This study | N/A |
| pHR_EF1a-->NLS-IFP-P2A-Ecad | This study | N/A |
| pHR_EF1a-->Pcad-IRES-mCherry | This study | N/A |
| **Antibodies** | | |
| PE/Cy7-conjugated anti-E-cadherin antibody | BioLegend | 147309 |
| **Chemicals, Enzymes, and other reagents** | | |
| DMEM | Nacalai Tesque | 08458-16 |
| Fetal bovine serum | Gibco | 10270-106 |
| 1x penicillin-streptomycin solution | Wako | 168-23191 |
| PBS | Wako | 049-29793 |
| PEI MAX | Polysciences | 24765 |
| hexadimethrine bromide | Sigma-Aldrich | H9268 |
| Agarose L | NIPPON GENE | 317-01182 |
| 60 mm non-treated culture dishes | Corning | 430589 |
| 12-well plates | Corning | 3513 |
| 96-well plates | Corning | 3595 |
| 96-well round-bottom ultra-low-attachment plate | Corning | 7007 |
| 384-well round-bottom ultra-low-attachment plate | Corning | 3830 |
| Two-well culture-insert | ibidi | 80209 |
| **Software** | | |
| FlowJo | BD Biosciences | |
| ImageJ | https://imagej.net/ij/ | |
| NIS-Elements | Nikon | |
| JOBS | Nikon | |
| OriginPro | OriginLab | |

| Reagent/Resource | Reference or Source | Identifier or Catalog Number |
|---|---|---|
| MATLAB | MathWorks | |
| Adobe After Effects | Adobe | |
| **Other** | | |
| SH800S | SONY | |
| SA3800 | SONY | |
| CytoFLEX S | Beckman Coulter | |
| Nikon Ti2 with Nikon AX-R | Nikon | |
| Nikon Ti2 with ANDOR Dragonfly | Nikon, ANDOR | |
| CFI Apochromat LWD Lambda S 20XC WI objective (0.95 NA) | Nikon | |
| CFI Plan Apochromat Lambda D10X objective (0.45 NA) | Nikon | |
| CFI Plan Apochromat Lambda D 20X objective (0.80 NA) | Nikon | |
| STXG-WSKMX-SET | TOKAI HIT | |

## Plasmid construct design for GFP-receiver cells

The chimeric protein anti-GFP synNotch receptor (Addgene, #162230) was constructed by fusing the following three proteins: anti-GFP nanobody LaG17 (Fridy et al, 2014) in the extracellular domain, mouse Notch1 (NM_008714) in the minimal regulatory region (Ile1427 to Arg 1752) in the transmembrane region, and TetR-VP64 in the intracellular domain (Morsut et al, 2016). To enable transmembrane protein expression, a CD8α signal sequence (MALPVTALLLPLALLLHAARP) was inserted at the N-terminus of synNotch receptor. In addition, a Myc tag (EQKLISEEDL) was fused to the N-terminus of LaG17 facilitate its detection in cell surface through immunostaining. The GFP-anchor protein (Addgene, #162225) consists of a CD8a signal sequence, an HA tag, an anti-GFP nanobody (LaG2), and a PDGFR transmembrane domain. These were then cloned into a modified pHR'SIN:CSW vector under the EF1α promoter. Human E-cadherin (NM_004360) was cloned under the control of the EF1α promoter to generate imC$^{Ecad}$ cells.

For inducible expression in response to synNotch stimuli, the following target genes were cloned under tetracycline-responsive element (TRE; TCCCTATCAGTGATAGAGA), fused to a minimal CMV promoter. TRE-->mCherry_PGK-->tagBFP (Addgene, #162231): The TRE promoter drives mCherry reporter expression, while the PGK promoter drives tagBFP as an integration marker. TRE-->E-cadherin-mCherry_PGK-->tagBFP: The TRE promoter drives the fusion of E-cadherin and mCherry, with the PGK promoter driving tagBFP as an integration marker. The C terminus of human E-cadherin was directly fused to the N terminus of mCherry.

## Plasmid construct design for GFP-secretor cells

Membrane-tethered GFP (Addgene, #79129) comprised an Ig kappa chain V-III region signal sequence (METDTLLLWVLLLWVPGSTGD), EGFP, and a PDGFR transmembrane domain. To enable protein secretion, secretory GFP was constructed by fusing the Gaussia luciferase signal sequence (MGVKVLFALICIAVAEA) to the N-terminus of EGFP. NLS (Nucleus Localizing Signal)-fused Infrared Fluorescence Protein (IFP2.0) connected to mouse P-cadherin via P2A ribosomal skipping sequence was cloned into the modified pHR'SIN:CSW vector under an EF1α promoter, allowing visualization of the GFP-secretor spheroid and preventing mixing with GFP-secretor cells and GFP-receiver cells in the spheroid culture.

## Plasmid construct design for other cadherin-expressing cells

For the Fig. 5A,B, NLS-fused IFP2.0 was connected to E-cadherin via the P2A ribosomal skipping sequence. To visualize the P-cadherin core spheroids in Fig. 5B, we fused P-cadherin to mCherry via the internal ribosome entry site (IRES). Cadherins were cloned under the control of the EF1α promoter in the modified pHRSIN:CSW vector.

## Cell culture

All cells were cultured in DMEM (Nacalai Tesque, #08458-16) supplemented with 10% fetal bovine serum (Gibco) and 1x penicillin-streptomycin solution (Wako, #168-23191) at 37 °C in humidified environment with 5% $CO_2$. The mouse fibroblast cell line L929 (RIKEN BRC, Cell Bank) was cultured on 60 mm TC-treated culture dishes (Greiner, #628160). The human erythroleukemic cell line K562 (ATCC, #CCL-243) was cultured in 60 mm non-treated culture dishes (Corning, #430589).

## Lentiviral transduction of L929 cell

All cells used in this study were stably transformed using lentiviral transduction. The cell lines were established as follows: (1) A lentivirus was generated by transfecting the pHR'SIN:CSW vector and the viral packaging plasmids pCMVdR8.91 and pMD2.G into HEK293T cells using PEI MAX (Polysciences, #24765). (2) $1.0 \times 10^5$ L929 cells were cultured with variable amounts of lentiviral supernatant and 10 μg/mL hexadimethrine bromide (Sigma-Aldrich, #H9268) in 12-well plates (Corning, #3513) for 48 h.

## Construction of synNotch-expressing cells (imC and iEcad-mC cells)

Following lentiviral infection with the anti-GFP synNotch expression plasmid and the inducible target gene plasmid, the cells were split into two wells in a 12-well plate and cultured overnight. Then $8.0 \times 10^5$ K562 cells expressing the membrane-tethered GFP ligand (K562$^{mGFP}$) were added to each well (Toda et al, 2018). After synNotch stimulation by K562$^{mGFP}$ cells for 24 h, we analyzed the basal/induced expression levels of reporter fluorescent proteins and sorted the target cells into 96-well plates (Corning, #3595) at a single cell per well using a cell sorter (SH800S, SONY). When the sorted cells reached confluence in each well of the 96-well plates, they were split into three 96-well plates. After two days of culture, $4.0 \times 10^4$ K562$^{mGFP}$ were added to each well of one of the 96 plates. Another plate was used as the control without stimuli, and the other plate was used to pick up the clones. After screening the clones using a cell analyzer (SA3800, SONY), we selected several

clones whose basal expression of the target gene was low, but induction was high. Flow cytometry data were analyzed using the FlowJo software.

## Pattern formation assay in SYMPLE3D

Organizer and receiver spheroids were prepared following these procedures (Fig. EV1B):

(1) GFP-secretor cells were plated at 50 cells/well in a 96-well round-bottom ultra-low-attachment plate (Corning, #7007) with 200 μL of medium. GFP-receiver cells were plated at 200 cells/well in 384-well round-bottom ultra-low-attachment plate (Corning, #3830) with 50 μL of medium. The plates were then centrifuged at $100 \times g$ for 2 min and incubated overnight.
(2) After spheroid formation, 160 μL of supernatant medium was removed from each well of the 96-well plate of secretor cells.
(3) 30 μL of medium containing a secretor spheroid, was transferred to each well of the 384-well receiver plate using tip-cut 200 μL pipette tip.
(4) Time-lapse imaging was started immediately, or snapshots were acquired after 48 h of incubation using a confocal microscope. Secretor cells were plated at varying densities of 25, 50, 100, and 200 cells/well to form secretor spheroids of variable sizes.

## Pattern formation assay in 2D culture system

A two-well culture-insert (ibidi, #80209) was placed in the center of each well in a 24-well plate. $1.6 \times 10^4$ GFP-secretor cells with 80 μL of medium were plated in the left side of the insert, while $8 \times 10^3$ GFP-receiver cells with 80 μL were plated in the right side. The outside of the insert was filled with $3 \times 10^4$ LaG2-anchor expressing cells as a sink. The plate was then centrifuged at $100 \times g$ for 2 min and incubated overnight. The left side of the insert (secretor cell region) was gently washed twice with 80 μL of medium. Then, after removing the insert and aspirating the supernatant, 800 μL of medium containing 1% Agarose L (NIPPON GENE, #317-01182) was slowly added to each well. This medium was prepared by mixing DMEM containing 20% FBS with PBS (Wako, #049-29793) containing 2% agarose at a 1:1 ratio. After leaving the plate at room temperature for 15 min for gelation, the plate was cultured for 48 h. The images were acquired by confocal microscopy.

## GFP titration assay

GFP receiver cells or parental L929 cells were plated at 100 cells/well in 40 μL of medium in a 384-well round-bottom ultra-low-attachment plate. The plate was then centrifuged at $100 \times g$ for 2 min and incubated for 24 h. Following the formation of spheroids, 40 μL of DMEM containing 10% FBS and various concentrations of purified GFP were added to each well. Spheroids were incubated for 48 h, followed by image acquisition.

## E-cadherin mixing assay

E-cadherin-high cells (L929 cells expressing E-cadherin under a EF1a promoter) were plated at 100 cells/well in 96-well round-

bottom ultra-low-attachment plate with 200 μL of medium. Equal number of L929 and imC or iEcad-mC cells (30 cells) were mixed in 30 μL medium and plated in a 384-well round-bottom ultra-low-attachment plate. The plate was then centrifuged at $100 \times g$ for 2 min and incubated for 24 h to form spheroids. We then transferred 30 μL of the medium containing E-cadherin high spheroid into a 384-well plate and added 30 μL of medium containing various concentrations of purified GFP. Time-lapse imaging started immediately after adding GFP or the spheroids were incubated for 48 h followed by image acquisition.

## Cell movement assay

To track cell movement inside E-cadherin-expressing spheroids, 199 E-cadherin-expressing cells and a single cell labeled with IFP (E-cadherin- and NLS-IFP2.0-expressing cells) were plated in 384-well round-bottom ultra-low-attachment plates. To compensate for spheroid movement and rotation during culture, a P-cadherin-expressing spheroid was placed next to an E-cadherin-expressing spheroid. The reference spheroid was formed by plating cells expressing P-cadherin and mCherry at a density of 50 cells/well in a 96-well round-bottom ultra-low-attachment plate with 200 μL of medium. The plates were then centrifuged at $100 \times g$ for 2 min and incubated overnight. After spheroid formation, 160 μL of supernatant medium was removed from each well, and 30 μL of medium containing the reference spheroid was transferred to each well of the 384-well plate containing E-cadherin-expressing cells using tip-cut 200 μL pipette tips. Time-lapse imaging was initiated immediately.

## Quantification of E-cadherin expression levels by immunostaining

iEcad-mC cells were plated at $0.5 \times 10^5$ cells/well in 12-well plates with 1 mL of medium. After a 24 h incubation, the medium was replaced with 1 mL of fresh medium containing various concentrations of GFP, and the cells were incubated for an additional 48 h. To detach the cells without proteolytic effects on E-cadherin, the cells were treated with 500 μL of 2 mM EDTA in PBS for 10 min. The cells were then stained with 1 μg/mL PE/Cy7-conjugated anti-E-cadherin antibody (BioLegend, #147309) in DMEM containing 10% FBS at 4 °C for 30 min. After the cell solution was washed twice with 500 μL of DMEM containing 10% FBS, the expression level was analyzed by flow cytometry (CytoFLEX S, Beckman Coulter), and the data were analyzed using the FlowJo software.

## Imaging

Snapshots, except for those in Fig. EV5B were acquired using a Nikon AX-R confocal microscope with a CFI Apochromat LWD Lambda S 20XC WI objective (0.95 NA) (Nikon). The snapshots for Fig. EV5B were acquired using an ANDOR Dragonfly confocal microscope with a CFI Plan Apochromat Lambda D10X objective (0.45 NA) (Nikon). Time-lapse images were captured using a CFI Plan Apochromat Lambda D 20X objective (0.80 NA) (Nikon). For this analysis, cells were kept in a precise incubation system (STXG-WSKMX-SET, TOKAI HIT) at 37 °C in a humidified atmosphere 5% $CO_2$. Using a Galvano scanner, images were acquired every 30 min for 48 h (97 images—Figs. 1C, 2C, 3B and Movies EV1–2). Using a resonant scanner, images were acquired every 10 min for

48 h (289 images—Fig. 4A,B and Movie EV3), every 20 min for 48 h (145 images—Fig. 5A and Movie EV4), or every 5 min for 24 h (289 images—Fig. 5B and Movie EV5). Microscopy and image acquisition were performed using NIS-Elements (Nikon). A high-content imaging system, JOBS (Nikon), was utilized to automatically scan the 384-well plates.

## Image analysis

We used ImageJ to quantitatively analyze the distribution of GFP, mCherry, and IFP (Figs. 1C,D, 2C,D, 3B,C,E and EV2A; Appendix Figs. S3C and S4B,C). The coordinates of the centroid of GFP-secretor cells were calculated after creating the mask by binarizing the IFP-2.0 fluorescence. Then, the rectangular ROI was automatically set from the coordinates to the end of a receiver spheroid.

To quantify the distribution of activated cells, mCherry fluorescence profiles were fitted to the Hill equation using OriginPro software:

$$y = START + (END - START)\frac{x^n}{k^n + x^n}$$

$k$: distance (μm) at half intensity of mCherry
$n$: Hill coefficient
START: Maximum value of mCherry intensity in the fitted curve
END: Minimum value of mCherry intensity in the fitted curve

The sharpness of mCherry decay was quantified using the Hill coefficient. A larger hill coefficient represents a sudden decrease in mCherry intensity, which indicates the segregation of the spheroid into active and inactive domains with sharper boundaries. The domain size was quantified using the $k$ value.

In Appendix Figs. S1A,B and S3D, rectangular ROIs with a width of 36 μm were vertically aligned from one end near the secretor spheroids to the other, and mCherry intensity profiles were calculated using ImageJ.

In Figs. 4C and EV3A–D, to precisely quantify the intensity of GFP and mCherry from spheroids, we first defined the ROIs of spheroid regions by selecting the regions over the threshold intensity of the lentiviral integration marker BFP, which included all imC, imC$^{Ecad}$, and iEcad-mC cells. The intensities of GFP and mCherry in the ROIs were measured. To quantify the compaction of the spheroids, spheroid areas were calculated using bright-field images corrected by rolling ball background subtraction using NIS-Elements. This quantification method for the compaction was also used in Fig. EV2B to quantify the spheroid areas.

In Appendix Fig. S2A,B, we generated a standard curve comparing the GFP intensity of spheroids with the GFP concentration in the medium. By taking spheroid images in the pattern formation and GFP titration assays with exactly the same settings of the confocal microscope, we estimated the GFP concentration inside the GFP-receiver spheroids.

In Fig. EV4A–C, to analyze the positions of receiver cells, we binarized the lentiviral integration marker, BFP, to measure the extent to which receiver cells were localized inside or outside an E-cadherin high-expressing spheroid. The region of the spheroid was defined by selecting regions with threshold intensity of IFP2.0 and smoothing the IFP2.0-positive ROI. We calculated the overlapping area between BFP and IFP as a mixed population of receiver cells in an E-cadherin high-expressing spheroid using NIS-Elements software.

In Fig. 5B, to record the dynamics of the distance between two daughter cells labeled with IFP2.0 expression after cell division, we detected the positions of daughter cells with binarized IFP2.0 in maximum intensity projection images. The distance between the center position of the ROIs was calculated using NIS-Elements software.

To create movies, time-lapse movies were edited using Adobe After Effects software.

## Mathematical modeling and simulation

### Dynamics of E-cadherin expression level
In our experimental setup, E-cadherin patterns were formed by the gradient of GFP along a specific axis, making the spatial profile along a single dimension the dominant factor, irrespective of the volumetric tissues. However, to accurately represent random cell movements and provide adequate degrees of freedom for cell motion, an additional dimension was required. Consequently, we developed a two-dimensional model, which effectively accounted for the phenomena in this study.

In the model, we considered receiver cells arranged in a hexagonal lattice, where each cell had six neighbors (except for boundary cells along the x-axis), with a typical length of 10 μm. We arranged 20 cells along the x-axis and 15 cells along the y-axis, and applied a periodic boundary condition for y-axis to ignore the impact of domain size. GFP source cells were positioned in the negative region of the x-axis, and the GFP spatial profile was formed along the x-axis through diffusion from these GFP sources at $x = 0$. We represented the spatial profile of GFP level $G$ using a simple exponential function:

$$G(x) = \alpha \exp(-\beta x), \tag{1}$$

where $\alpha$ controls the source level and $\beta$ controls the spatial range. For our standard parameter set, we chose $\alpha = 1$ and $\beta = 0.05$ to mimic the spatial profile of GFP in the receiver domain.

E-cadherin expression is induced by GFP and undergoes self-degradation with a turnover rate $\tau$. To model the dynamics of E-cadherin expression level $C$, we employed a simple linear function:

$$dC/dt = \tau(sf(G) - C), \tag{2}$$

where $s$ is induction coefficient, and we set $s = 1$ unless otherwise specified. The function governing the influence of GFP on the induction of E-cadherin expression level was determined from our experimental measurements in a normalized form and represented by the best-fitted quintic function: $f(G) = 1.93G^5 - 10.36G^4 + 18.43G^3 - 14.5G^2 + 5.48G + 0.01$ for $G \in [0, 1]$ (Appendix Fig. S5A). For the tunability analysis of E-cadherin domain size, we set $f(G) = 1$ for $G > 0$ when we set $\alpha > 1$ (Appendix Fig. S5C). The solution of Eq. (2) is expressed as:

$$C(t,x) = sf(G) + (C_0 - sf(G)) \exp(-\tau t), \tag{3}$$

where $C_0$ represents the initial value of E-cadherin expression level. As the timescale of this dynamics is characterized by $\tau^{-1}\ln\{2|sf(G) - C_0|/(sf(G) + C_0)\}$, a larger $\tau$ results in a faster equilibrium in the system as depicted in Fig. 6H.

## Dynamics of cells

E-cadherin-expressing cells are spatially arranged through cell sorting based on differential E-cadherin expression between neighboring cells. To model this process, we defined the differential adhesion energy (DAE) $E$ as follows:

$$E = \sum_{i}^{N} \sum_{j}^{n_i} \Delta C_{ij}^2 / n_i, \tag{4}$$

where $i$ and $j$ represent cell indices, $N$ is the total number of cells, $n_i$ denotes the number of neighboring cells of cell $i$, and $\Delta C_{ij}$ represents the function expressing differential E-cadherin expression of cell $i$ and its neighbor $j$.

As described in the main text, effective E-cadherin-based cell sorting becomes disabled when the E-cadherin level exceeds a threshold value $\theta_i$ with the cell index $i$. For the cell-to-cell variance in the threshold values (Appendix Fig. S5B), we used a normal distribution with parameters mean $\theta$ and standard deviation σ. To incorporate this saturation regime, we define $\Delta C_{ij}$ as follows:

$$\Delta C_{ij} = 0 \qquad \text{if } C_i > \theta \wedge C_{i,j} > \theta,$$

$$\Delta C_{ij} = C_i - C_{i,j} \quad \text{otherwise}, \tag{5}$$

where $C_i$ and $C_{i,j}$ denote E-cadherin expression level in cell $i$ and that in its neighbor cell $j$, respectively. Based on our experimental measurements, we set the threshold $\theta_i = 0.3$ for all cells as the standard unless stated otherwise. In the no-saturation regime, we defined $\Delta C_{ij} = C_i - C_{i,j}$, regardless of values in $C_i$ and $C_{i,j}$.

Cell movement in the model occurs stochastically using a lattice-based Monte Carlo method, aiming to lower the system's energy. In this process, a randomly selected cell $i$ is swapped with one of its neighboring cells, $j$, which is also chosen randomly. The transition is determined by evaluating the change in energy $\Delta E$ associated with this replacement, where $\Delta E$ is calculated as $\Delta E = E_{\text{after}} - E_{\text{before}}$. The replacement takes place stochastically according to the Boltzmann acceptance function $\exp(-\xi \Delta E)$ when $\Delta E > 0$, where $\xi$ is a parameter controlling the relative strength of DAE-based cell sorting versus random cell movement. When $\Delta E \leq 0$, the replacement occurs deterministically. Through repeated trials of such cell swapping, the system gradually transitions towards a lower-energy state. We set $\xi = 100$ as a standard parameter value to capture both the DAE-based cell sorting and random cell movements.

We considered the total number of cells ($N = 300$) as a unit of the simulation step, equivalent to one Monte Carlo step (mcs), and each mcs corresponds to 1 h. E-cadherin expression level used in the calculation of DAE was updated every mcs to reflect 1 h of dynamics from Eq. (3), leading to

$$C(mcs + 1) = sf(G) + (C(mcs) - sf(G)) \exp(-\tau). \tag{6}$$

for positive integers of mcs. As for the initial condition, we set the E-cadherin expression profile $C(mcs = 0) = 0$ for all cells. Altogether, this allowed us to capture the temporal changes in E-cadherin expression and cell dynamics in the simulation.

## Simulations for EACs

We assumed that EACs were generated randomly in the E-cadherin-negative region every 5 h, starting from $t = 24$ h until $t = 100$ h. The absorption probability of these EACs was assessed within a time frame of 20 h after their generation to determine whether they were absorbed into the E-cadherin-positive domain. To model EACs with the high sensitivity, we set the induction coefficient as $s = 10$. However, the choice of this value, when $s > 1$, did not result in any qualitative differences in the results. For the case of no cell sorting, we set $\xi = 0$. In cases involving cell sorting, we explored different values of $\xi$, but no qualitative differences in the overall trends were observed in our simulations (Appendix Fig. S5D).

## Data availability

The datasets (and computer code) produced in this study are available in the following databases: Images: BioImage Archive S-BIAD1203 (https://doi.org/10.6019/S-BIAD1203). Computational codes: GitHub (https://github.com/tsuyoshihirashima/morphogen-sorting) and Zenodo (https://zenodo.org/doi/10.5281/zenodo.13312412).

The source data of this paper are collected in the following database record: biostudies:S-SCDT-10_1038-S44319-024-00261-z.

## Peer review information

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

## Acknowledgements

We thank T. Ishitani, S. Okuda, S. Arai and members of the Toda laboratory for discussion and assistance. This work was supported by the Japan Science and Technology Agency (JST), PRESTO Grant No. JPMJPR2147; the Japan Society for the Promotion of Science (JSPS), KAKENHI Grant No. 20K15828, and 21H05291 to ST, and 21H05290 to TH; and the Japan Agency for Medical Research and Development (AMED), Grant No. 22bm0704048h0003, Japan; the Senri Life Science Foundation, Japan; the Kato Memorial Bioscience Foundation, Japan; The Kao Foundation for Arts and Sciences, Japan to ST; and World Premier International Research Center Initiative (WPI), Ministry of Education, Culture, Sports, Science and Technology (MEXT), Japan to KM and ST; and the Mechanobiology Institute (MBI) at the National University of Singapore (NUS) funded through the National Research Foundation, Singapore and the Ministry of Education, Singapore under the Research Centre of Excellence programme, and by the Department of Physiology at NUS to TH. KM is supported by the Yoshida Scholarship Foundation PhD fellowship.

## Author contributions

**Kosuke Mizuno**: Conceptualization; Data curation; Formal analysis; Funding acquisition; Validation; Investigation; Visualization; Methodology; Writing—original draft; Writing—review and editing. **Tsuyoshi Hirashima**: Conceptualization; Resources; Data curation; Software; Formal analysis; Funding acquisition; Validation; Investigation; Visualization; Methodology; Writing—review and editing. **Satoshi Toda**: Conceptualization; Resources; Supervision; Funding acquisition; Validation; Investigation; Visualization; Methodology; Writing—original draft; Project administration; Writing—review and editing.

Source data underlying figure panels in this paper may have individual authorship assigned. Where available, figure panel/source data authorship is listed in the following database record: biostudies:S-SCDT-10_1038-S44319-024-00261-z.

## Disclosure and competing interests statement

Satoshi Toda is an inventor on a patent for synthetic Notch receptors (Patent No.: US 10,590,182 B2) held by the Regents of the University of California, which is licensed to Gilead. The remaining authors declare no competing interests.

# Expanded View Figures

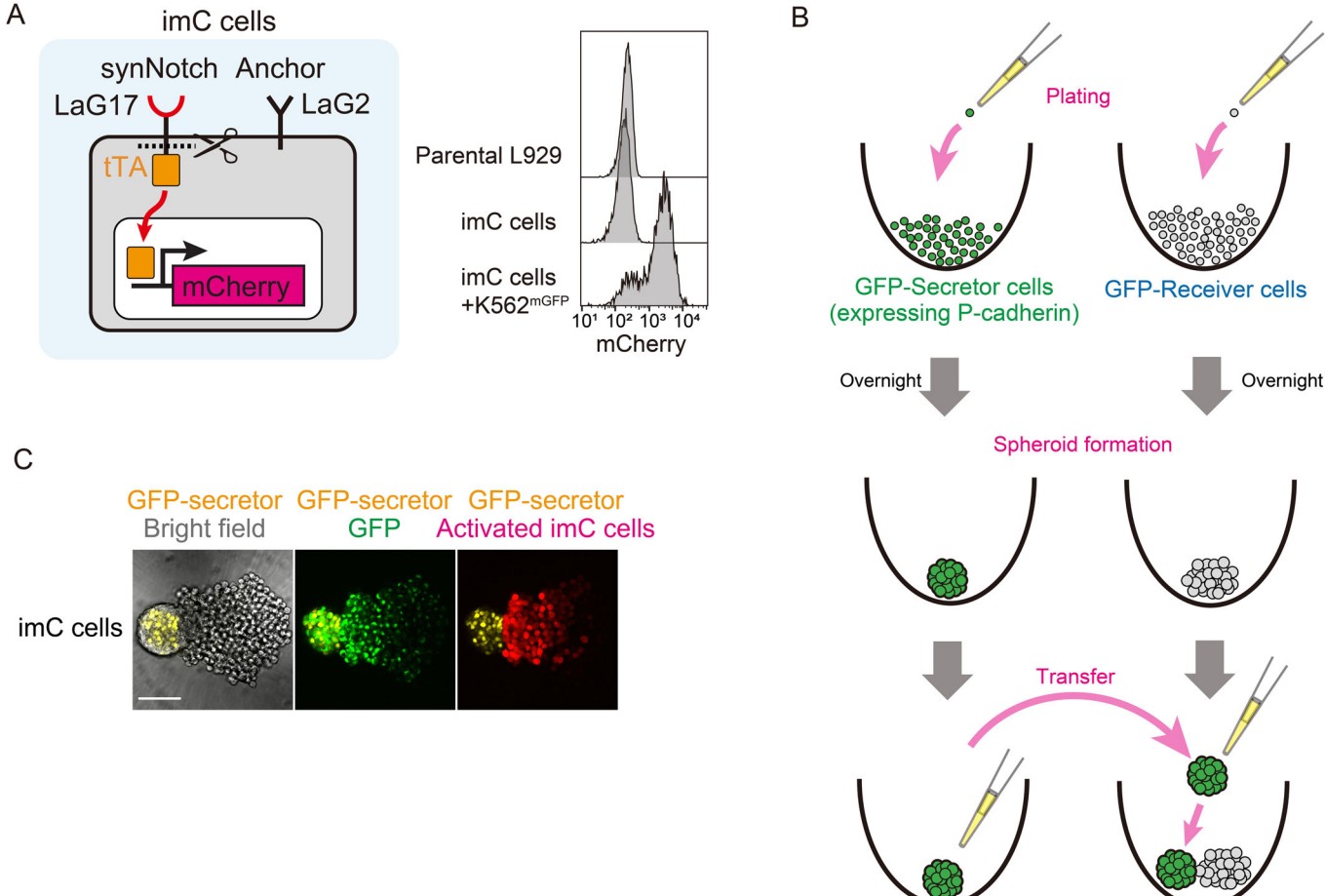

**Figure EV1. Experimental setup of 3D synthetic morphogen system.**

(**A**) Establishment of imC cells: imC cells express the GFP anchor protein, which captures GFP on the cell surface, and the anti-GFP synNotch receptor, which induces the mCherry reporter upon recognizing the captured GFP. The GFP-anchor protein and anti-GFP synNotch receptor utilize different nanobodies (LaG2 and LaG17, respectively) that recognize different epitopes on GFP. To confirm the induction of mCherry reporter by the anti-GFP synNotch receptor, imC cells were stimulated with K562 cells expressing membrane-tethered GFP (K562$^{mGFP}$) (Toda et al, 2018) and analyzed using flow cytometry. (**B**) Co-culture of GFP-secretor and GFP-receiver spheroids. GFP-secretor and receiver cells were separately plated and cultured overnight to form GFP-secretor and GFP-receiver spheroids, respectively. The GFP-secretor spheroid was then transferred to a well containing a GFP-receiver spheroid to initiate co-culture. (**C**) Synthetic gradient formation by GFP morphogens in imC cell spheroids. Bright-field, GFP, and mCherry channels of the synthetic gradient at 48 h. GFP-secretor cells labeled by nucleus-localized IFP2.0, which is overlaid with a yellow pseudocolor. Scale bar: 100 μm.

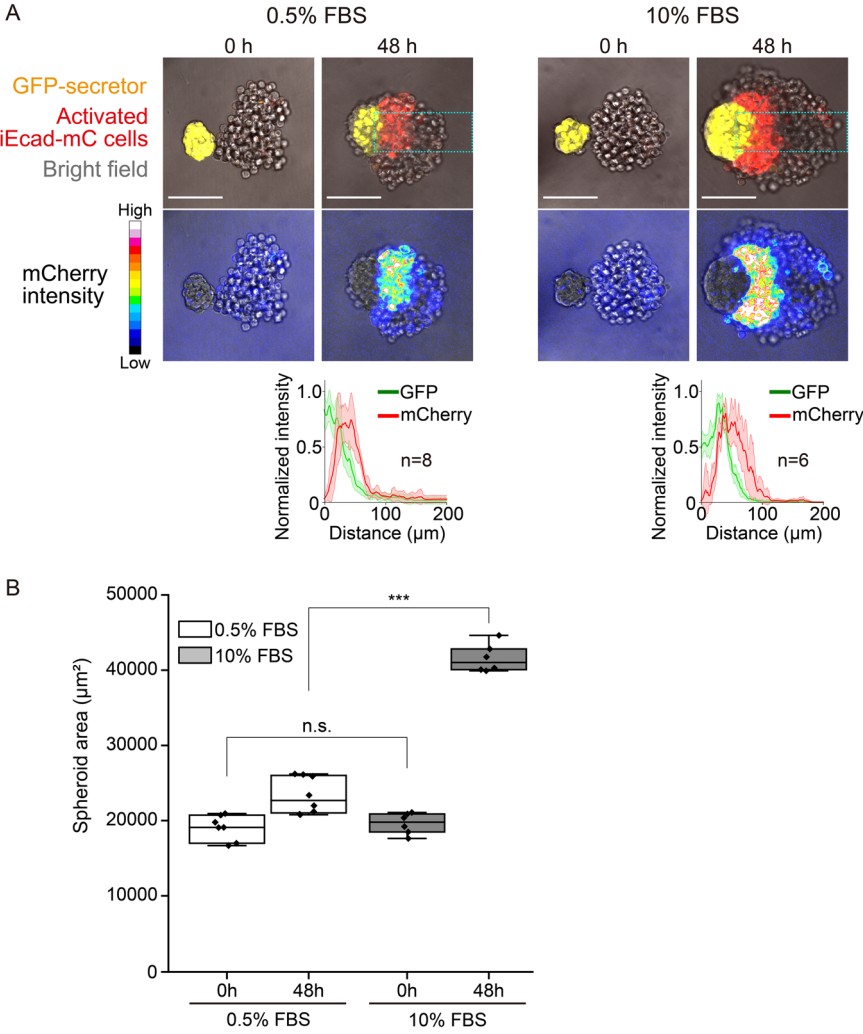

**Figure EV2. Robust pattern formation in a low serum condition.**

(A) Confocal images of iEcad-mC spheroids incubated in DMEM containing 0.5% FBS (left) or 10% FBS (right). GFP-secretor spheroids with 50 GFP-secretor cells and GFP-receiver spheroids with 200 imC or iEcad-mC cells were separately formed and co-cultured from time 0 h. Spheroid images acquired at 0 and 48 h using a confocal microscope. GFP secretor cells express nucleus-localized IFP2.0, shown as a yellow pseudocolor. Scale bar: 100 μm. mCherry distributions are visualized by 16 pseudocolors in the bottom images. Fluorescence intensity profiles within rectangular regions (green line: GFP, red line: mCherry) are overlaid at the bottom of the images. The means are indicated with bold lines (green line: GFP, red line: mCherry). Shaded areas in the graph represent ± SD. 0.5% FBS: $n = 8$, 10% FBS: $n = 6$. (B) The graph indicated the spheroid size at 0 and 48 h when incubated with 0.5% FBS (white box) and 10% FBS (gray box). The spheroid size slightly increased in a 0.5% FBS condition after 48 h of incubation, whereas it increased more than double in a 10% FBS concentration. The boxes represent the group median and interquartile range (25th–75th percentiles). The whiskers extend to the minimum and maximum data points. Differences between two groups were determined using Welch's t-test with *** for $P < 0.001$ and n.s. (non-significant) for $P > 0.1$ ($P = 0.51$ between 0.5% FBS at 0 h and 10% FBS at 0 h. $P = 2.2 \times 10^{-9}$ between 0.5% FBS at 48 h and 10% FBS at 48 h). The same samples were observed and measured at 0 and 48 h in each condition ($n = 8$ for 0.5% FBS, $n = 6$ for 10% FBS).

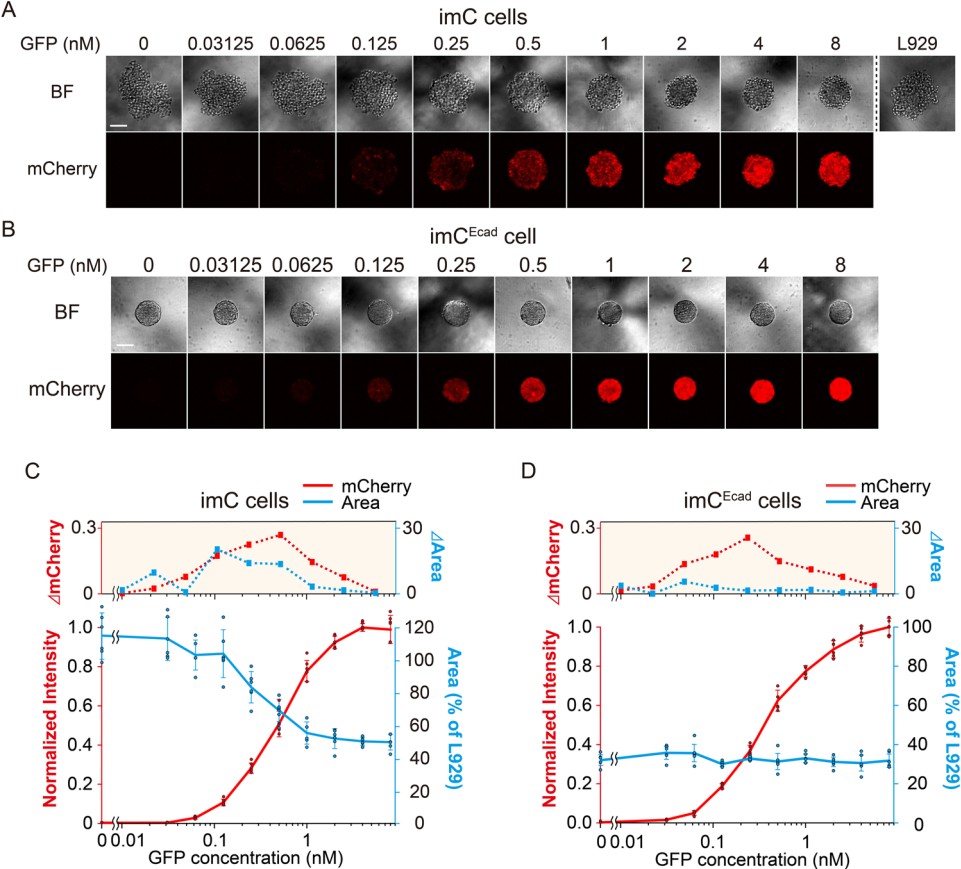

**Figure EV3. mCherry induction and spheroid shape change of imC and imC^Ecad spheroids with variable GFP concentrations.**

(A, B) Systematic analysis of the relationship between mCherry induction and spheroid shape of imC cells (A) and imC^Ecad cells (B). Spheroid images upon GFP concentrations ranging 0 to 8 nM. Spheroid of parental L929 is shown as control. Scale bar:100 μm. (C, D) Quantification of mCherry induction and spheroid compaction of the imC cells (C) and imC^Ecad cells (D). Rate of spheroid compaction quantified by comparison of the cross-sectional area of the spheroids with that of parental L929 spheroids using bright-field images. In the right graph, the left and right numbers indicate normalized fluorescence intensity of E-cadherin-mCherry and the percentage of the spheroid area relative to L929 spheroids, respectively. The difference of values between adjacent GFP concentrations is plotted in the top graph. The imC spheroids gradually shrunk in response to GFP, while the mCherry induction level was increased proportionally to the increase in GFP concentration. However, since each cell shape was round and did not form a smooth spheroid surface, this morphological change was not defined as compaction as it was caused by stronger binding to each other with higher concentration of GFP, which bridged imC cells through the trans-interaction of GFP-anchor protein/GFP/anti-GFP synNotch receptor. On the other hand, the imC^Ecad cell spheroids exhibited a compact morphology due to E-cadherin expression. The addition of GFP had no effect on spheroid compaction, while the induction level of mCherry reporter increased. The GFP concentrations at the curve rising (i.e., sensitivity to GFP) of imC cells and imC^Ecad cells were similar, indicating that E-cadherin expression does not influence anti-GFP synNotch activation. Experiments were performed in 4–6 replicates, and data are plotted mean ± SD.

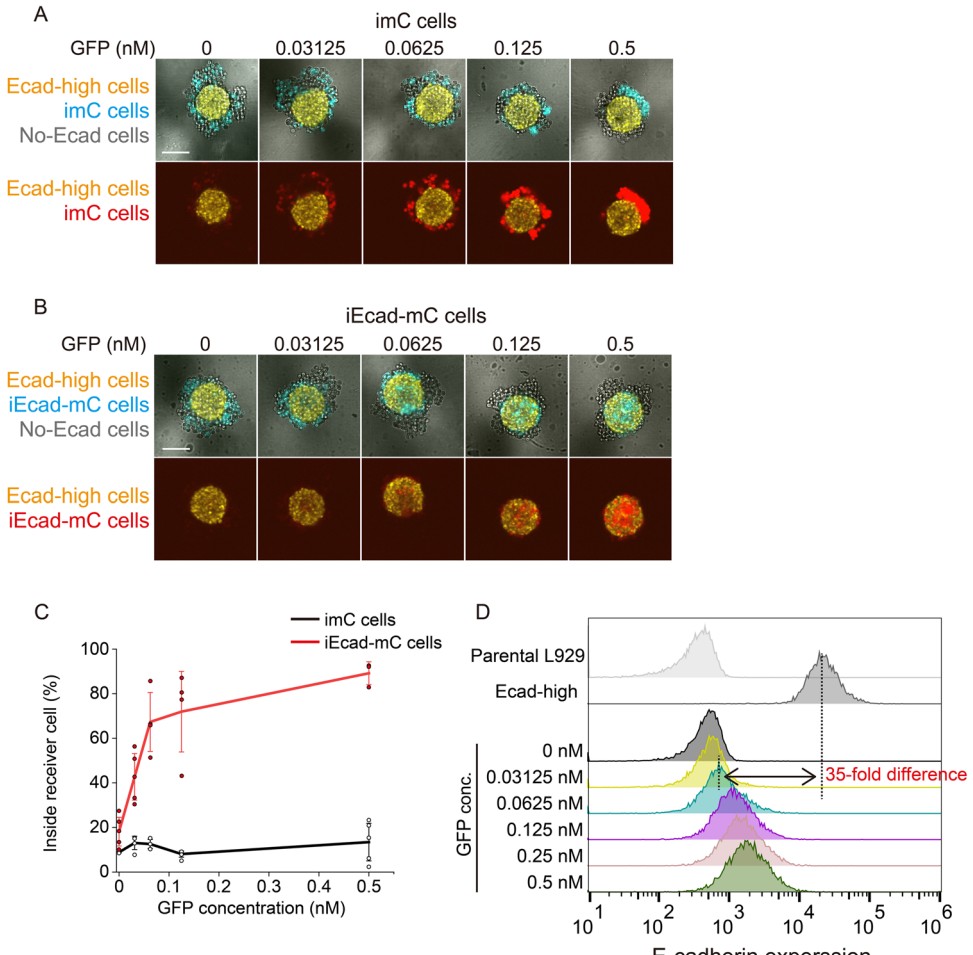

**Figure EV4. Mixing of E-cadherin high-expressing cells and iEcad-mC cells.**

(A, B) Activated iEcad-mC cells above a threshold level mixed well with E-cadherin high-expressing cells (Ecad-high cells). Spheroids composed of 100 Ecad-high cells were mixed with spheroids composed by combining 30 imC cells and 30 parental L929 cells (no-Ecad cells) (A), or 30 iEcad-mC cells and 30 no-Ecad cells (B) and stimulated with variable GFP concentrations. Ecad-high cells, imC/iEcad-mC cells, and mCherry/E-cadherin-mCherry are labeled with yellow, cyan, and red, respectively. Scale bar: 100 μm. (C) Percentage of imC or iEcad-mC cells mixed with Ecad-high cells. Results from 3–5 replicates are plotted as mean ± SD. (D) E-cadherin expression levels of parental L929 cells, Ecad-high cells, and iEcad-mC cells in response to varying GFP concentrations were analyzed with immunostaining using an anti-E-cadherin antibody. The mean value difference between Ecad-high and iEcad-mC cells at 0.0625 nM GFP (dashed lines) is around 35-fold.

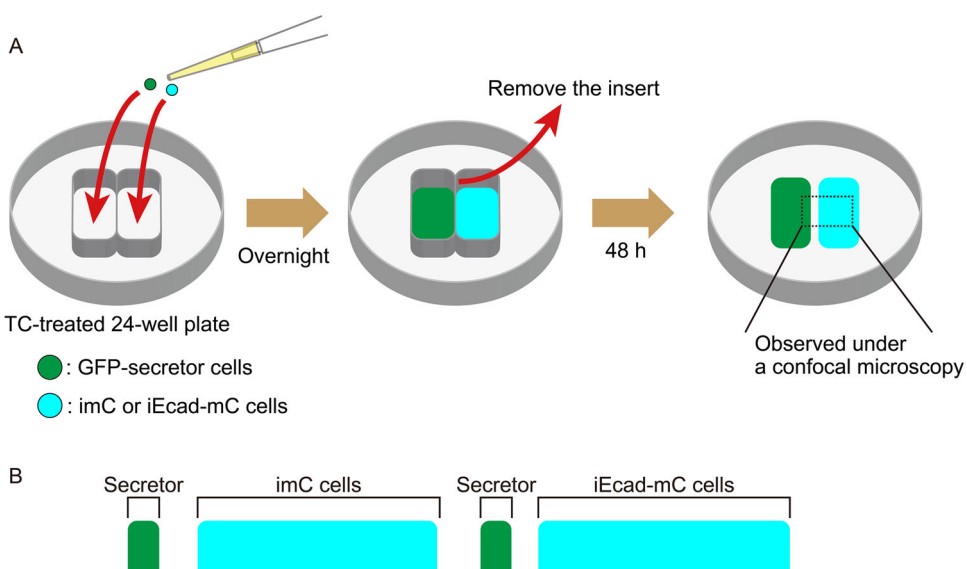

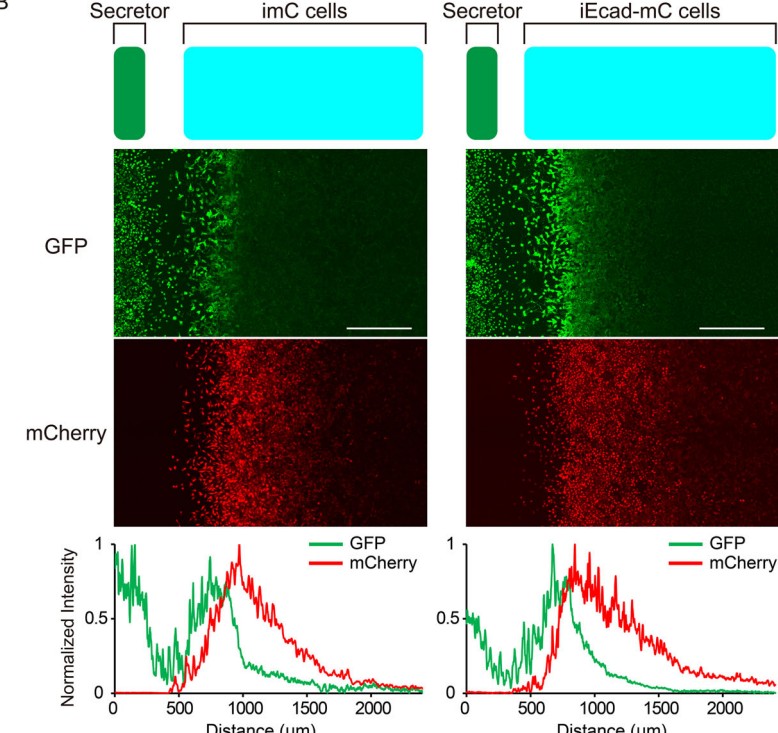

**Figure EV5. Pattern formation assay in a 2D culture condition.**

(A) Experimental setup of pattern formation assay in a 2D culture system using a two-well culture-insert. $1.6 \times 10^4$ GFP-secretor cells were plated in the left side of the insert, while $8 \times 10^3$ GFP-receiver cells were plated in the right side. The insert was removed after the cells were incubated overnight. The interaction between two regions was observed by confocal microscopy after another 48 h. (B) The mCherry distributions of both imC and iEcad-mC cells showed a gradient pattern. GFP-secretor cells were localized to a narrow area on the left. GFP-receiver cells were localized to the area to the right of the GFP-secretor cells, depicted in green and cyan in the upper cartoon above the confocal images, respectively. Scale bar: 500 μm. Fluorescence intensity profiles (green line: GFP, red line: mCherry) are overlaid below.

