## [Peer Review File · EMBO Reports]

Robust tissue pattern formation by coupling morphogen signal and cell adhesion

Kosuke Mizuno, Tsuyoshi Hirashima, and Satoshi Toda

Corresponding author(s): Satoshi Toda (satoshi.toda@protein.osaka-u.ac.jp), Tsuyoshi Hirashima (thira@nus.edu.sg)

Review Timeline:

Submission Date:	14th Mar 24
Editorial Decision:	15th Mar 24
Revision Received:	15th Jun 24
Editorial Decision:	13th Aug 24
Revision Received:	23rd Aug 24
Accepted:	29th Aug 24

Editor: *Martina Rembold*

Transaction Report: A revised version of this manuscript was transferred to EMBO reports following peer review at the EMBO Journal.

Referee #1:

The authors provide evidence - using cell organoids - that E-cadherin-mediated cell sorting enables robust tissue patterning in the presence of noisy morphogen gradients.

The observation that cadherin-mediated cell sorting plays a decisive role in the process where morphogen gradients are converted into distinct tissue domains is not entirely novel. However, a systematic analysis of this mechanism using a synthetic morphogen system in 3D spheroids has not yet been undertaken.

While the manuscript is well written and the experimental results are clearly presented, several questions remain as to the correct interpretation of the results.

1. E-cadherin has both a mechanical (adhesive) and signaling function. From the data presented, it is not clear how E-cadherin triggers the observed cell sorting - do cells display random motion and sort by differential adhesion, or does E-cadherin signaling affect other cellular properties such as motility, protrusion formation etc. This can be addressed by using truncated/signaling defective forms of E-cadherin and a more in-depth analysis of cell behavior during the sorting process.
2. A prominent feature of E-cadherin-mediated cell-cell adhesion is its mechanosensitivity. Is there any evidence of mechanosensitivity playing a role during the sorting process? This can be easily analyzed by monitoring Vinculin recruitment to adhesion sites of alpha-catenin unfolding using suitable antibodies.
3. One notable weakness of the synthetic assay system used is that the default cell-cell adhesion (in the absence of GFP signaling) is unclear and thus the cause for differential adhesion unknown. It would be much nicer to use a dual adhesion system where cells can express both e-cadherin and n-cadherin and GFP signaling would change the ratio of their expression. I realize that this might constitute some more work, but it would considerably improve this study.
4. A critical prerequisite for cell sorting is cell motility (random mixing), which is particularly important for eliminating ectopically activated cells. It might be quite revealing to modulate the level of cell motility in the assay system to evaluate the minimal level of cell motility as a function of the e-cadherin gradient.

Taken together, this is an interesting manuscript, which needs a little more work before it is conclusive.

Referee #2:

Recently, there has been considerable interest in development synthetic morphogen systems, and a number of studies have developed systems in which secreted GFP can serve as a morphogen to

receiving cells. In these systems, the GFP is captured by an anchor molecule and then presented to the neighboring cell which contains a nanobody for GFP coupled with the synNotch system. In this manuscript, Mizuno et al. develop a synthetic system for tissue patterning by combining one such synthetic morphogen system with receiver cells which express E-Cadherin in response to receiving the GFP signal. The authors show that coupling these two systems produces sharper boundaries in the receiving cells, and that this system can be used to create patterns and to study the mechanisms of robust patterning. Overall the experiments are well described and carried out, however, there are a number of conceptual and technical issues with the study that I describe below:

Major points:

1. It is clear from examining the data in Figs 1-3, that the expression of the cadherin molecule and the subsequent cell rearrangements that compactify the receiving domain also modify the gradient of GFP. This is true in both the constitutively expressed cadherin (Fig 2) and the cadherin induced by GFP (Fig 3), although the effects do not appear the same in these two cases. This will in turn feedback on the expression of the mCherry fluorescent protein and E-cadherin, and so some of the enhanced sharpness may come from sharpening the gradient, not only from cell rearrangements.
2. This effect is not captured in the mathematical model which uses a simple exponential to describe the GFP gradient. Thus, the model will not be able to accurately capture the different gradient shapes and relate the sharpness of the gradient to that of the induced pattern.
3. The authors do not consider the effects of cell division. It may be that much of the GFP is trapped upon entering the receiving cells but then spreads through cell division creating a gradient. Similarly, mCherry may be induced along the border and then the responsive domain may spread through growth rather than signal reception. This latter point could be addressed using a Notch inhibitor like DAPT to block signal reception at different times as the authors did in their previous study (Toda et al Science 2018). The effects of division are also not captured in the modeling but may play an important role in shaping the results.
4. The quantifications and statistics are insufficient throughout. The methods indicate that the authors simply manually take a line profile through the aggregate using ImageJ. In each case, it appears that a single such profile is presented. The authors should quantify several aggregates for each case using automated image analysis rather than manual line profiles. The results should be presented with averages over aggregates as well as error bars and differences between conditions (i.e. with cadherin and without) should be evaluated statistically.
5. The mathematical model should be carefully compared to the data. Without doing this, the model doesn't add much to the paper.
6. The advance of the paper appears to be limited. My main expertise is in developmental biology and nothing truly new about developmental systems has been learned. Natural systems that use the principles here have already been described and are cited by the authors. I am not a synthetic biologist, however, the advance from this perspective appears to be limited as well. Both the systems using GFP as a synthetic morphogen and placing cadherins downstream of synNotch have been described previously, including by these same authors. The main new aspect of the paper is coupling these existing systems.

Minor points:

1. It isn't clear how well this system will translate to other cell types, particularly epithelial cells which may already express cadherin molecules. The authors state in lines 325-327 that these are potential applications without explaining how this would work.

2. In figures 1C, 2C, 3B, the authors should find a different way to label the GFP-secretors (IFP2.0 label); at first glance, it appears to show high mCherry intensity in secretor cells

Referee #3:

This is an interesting work using synthetic biology to model tissue patterning. The authors found the coupling of a GFP-based synthetic morphogen gradient with E-cadherin-based cell adhesion can improve the robustness of tissue patterning. The crosstalk between morphogen gradient and E-cadherin revealed a surprising switch-like effect. The cells exhibit compaction when the E-cadherin level crosses a critical threshold. This switch-like behavior led to a sharp boundary between two domains with uniformly high E-cadherin expression and low E-cadherin expression.

The principles revealed by this work are interesting to the general audience of developmental biologists, adhesion cell biologists, and synthetic biologists. The idea that coupling morphogen signaling and cell adhesion can enhance precision in tissue patterning is well-established and not new. The SYMPLE3D platform offered a new synthetic system that confirms this established concept. The artificial and tunable components of SYMPLE3D will be useful for future studies aiming to dissect various crosstalk mechanisms between morphogen signaling and adhesion. In my opinion, the most significant conceptual advance is the idea that a switch-like behavior of cell compaction in morphogen-driven cadherin expression can generate two spatial domains with a sharp boundary.

I am generally positive about this work. The experiments are well done and well-explained in the manuscript. I only have some suggestions for clarification and extra information that might be beneficial for readers.

What cadherins do L929 cells express? It appears that L929 cells have some basal adhesiveness. For example, the P-cadherin expressing GFP secretor forms significant adhesion with imC cells without the manipulation of E-cadherin. What adhesion molecules hold these cells together? A more thorough analysis of cadherins expressed in the L929 cells would help interpretation of later results (e.g., the threshold of E-cadherin over-expression)

Choice of P-cadherin & E-cadherin: Why did the authors choose to over-express P-cadherin in GFP secretors and E-cadherin in imC cells? Is there a specific logic behind it? As many cadherins form heterophilic adhesion, are P-cadherin and E-cadherin chosen because they do not form heterophilic adhesion? Are there previous work to support this? If there are, citations would be needed. There are likely more, but one related work is Min Bao, ..., Magdalena Zernicka-Goetz Nature Cell Biology 2022, where P-cadherin and E-cadherin are utilized to improve sorting of stem cell derived germ layer progenitors.

Switch-like compaction on induced E-cadherin: Figures 4 and 5 are very interesting, and potentially challenge the classic differential adhesion hypothesis (DAH). The DAH suggests cells can sort based on quantitative differences of cadherins. Here the result suggests the sorting behavior is "all-or-none". The cells either express more E-cadherin than the threshold and mix well with the other E-cad-high cells or vice versa. I would love to see more conditions in Fig 5A. What range of E-cadherin expression can E-cad-high cells mix well? (10-fold-difference? 100-fold-difference?) Can the author (1) estimate how much difference in E-cad level between Ecad-high cells vs. iEcad-mC cells in Fig 5A and (2) test how much difference in E-cad level can cells still sort to the same E-cad-high compartment? Is there no difference or will there be a bias (say E-cad-super-high cells still cluster together more than E-cad-medium-high cells) when the fold-difference is higher?

Mathematical model: The mathematical model aims to capture the sorting dynamics under GFP-induced E-cadherin gradient. While no mathematical model is perfect, I do believe this model is on the simpler side, almost too simple that intuition (like Fig 7) can already predict the simulation outcome. The author set up a relatively artificial scheme, where there is zero differential adhesion energy when E-cadherin in both neighbors exceeds a threshold. The y direction follows a periodic boundary condition (so no change of spheroid shape). The E-cadherin gradient naturally follows the quintic function and the sorting outcome is expected. This model does not consider contact size fluctuations (dynamic increase or shrink of cell contacts) and the complex neighbor exchange dynamics as it assumes the edge length is constant and the cells are simply swapping positions. By this simplification, the model becomes confirmatory of existing intuition, rather than generating new insights. The added value of Fig 6 is therefore not too different from a conceptual cartoon (Fig 7).

As a suggestion, a Cellular Potts Model (e.g., CompuCell3D) can more accurately capture cell contact dynamics and the evolution of domain shape (although the model has its inaccuracies too). Given that the experimental part is quite strong, I would not insist the author come up with a more sophisticated modeling scheme than the current version (But they should in future works!) Instead, the author should acknowledge all the limitations of the model (e.g., it is not modeling the whole cell but just considering the probability of cell switching position based on an artificial neighbor energy difference, it does not consider edge length fluctuation, the boundary condition does not allow the spheroid or the E-cad high domain to change shape in Y, etc.)

I do have one suggested addition to the math modeling. Does the author consider scenarios where the sorting threshold (θ) is fluctuating? As the model stands, all cells have identical thresholds to determine whether they sort or not. What if this θ is variable (which it most likely does)? How does this change the boundary between the E-cadherin high and low domains, and the absorption of EACs?

“EMBO Reports would require further information on the switch-like behavior by testing more experimental conditions (referee #3), further data on the general adhesiveness/Cadherin expression of the cells (referee #1 and #3), a refinement of the mathematical model, as outlined by referee #2 and #3. All control experiments and concerns regarding quantification and statistics need to be addressed. Testing the adhesive or signaling role of E-cadherin is recommended (referee #1). Mechanosensitivity and cell motility (referee #1, point 2 and 4) would certainly further strengthen the study and we encourage but do not request these experiments.”

Revision points:

further information on the switch-like behavior by testing more experimental conditions (referee #3)

As suggested by referee #3, we will quantify the E-cadherin expression levels by antibody staining to figure out the fold-difference in E-cadherin expression between each cell condition. We will check if there is no difference or a bias (say E-cad-super-high cells still cluster together more than E-cad-medium-high cells) when the fold-difference is higher.

further data on the general adhesiveness/Cadherin expression of the cells (referee #1 and #3)

We will find the RNAseq or microarray transcriptome data of L929 cells from a published database to check if L929 cells express endogenous cadherins and other adhesion molecules.

a refinement of the mathematical model, as outlined by referee #2 and #3

The referee #2 said, “the expression of the cadherin molecule and the subsequent cell rearrangements that compactify the receiving domain also modify the gradient of GFP” and suggested the refinement of mathematical model based on the idea. We agree that E-cadherin expression compactified the receiving domain, but there are no experimental data supporting that E-cadherin expression modified GFP gradient. Though the gradient range was shorter by E-cadherin-based compaction comparing Fig. 1 (without E-cadherin) and Fig. 2 (with E-cadherin), the gradient range against the spheroid diameter is similar, suggesting that there are similar number of cells inside the gradient with/without E-cadherin. The gradient range could be determined by the balance of GFP production by secretor cells and GFP consumption by receiver cells. Given this, when the spheroid undergoes compaction, the absolute gradient range becomes shorter due to the scaling effect; however, the gradient range against spheroid diameter, or the number of cells inside the gradient do not change.

We plan to add further discussion in details to avoid the misunderstanding of gradient regulation. Then we plan to refine our mathematical modeling according to the referee comments.

All control experiments and concerns regarding quantification and statistics

We will redo the quantification of mCherry reporter profiles in the receiver spheroids within the ROI set in an automated manner. We will show the averages of mCherry reporter profiles over aggregates as well as error bars and the differences between conditions with/without E-cadherin induction.

Testing the adhesive or signaling role of E-cadherin is recommended (referee #1)

We already have the cells expressing a full length of E-cadherin or a truncated E-cadherin that lacks the intracellular domain. To examine the role of E-cadherin intracellular domain and intracellular signaling during cell sorting, we will compare the cell sorting process in the mixed culture of wildtype cells and E-cadherin-expressing cells or truncated E-cadherin-expressing cells.

Dear Satoshi,

Thank you for the transfer of your research manuscript to our journal. As my colleague from The EMBO Journal, Ieva Gailite, outlined, we would like to invite you to revise your study for potential publication in EMBO Reports. We would require further information on the switch-like behavior by testing more experimental conditions (referee #3), further data on the general adhesiveness/Cadherin expression of the cells (referee #1 and #3), a refinement of the mathematical model, as outlined by referee #2 and #3. All control experiments and concerns regarding quantification and statistics need to be addressed. Testing the adhesive or signaling role of E-cadherin is recommended (referee #1). Mechanosensitivity and cell motility (referee #1, point 2 and 4) would certainly further strengthen the study and we encourage but do not request these experiments.

We have already discussed these revisions further and agreed that you revise your study along the revision plan you sent. In the interest of protecting the conceptual advance provided by the work, we recommend a revision within 3 months (June 15). That said, please contact me if you require more time to complete the revisions, as discussed.

Please address all referee concerns in a complete point-by-point response. Acceptance of the manuscript will depend on a positive outcome of a second round of review. It is EMBO Reports policy to allow a single round of revision only and acceptance or rejection of the manuscript will therefore depend on the completeness of your responses included in the next, final version of the manuscript.

*****IMPORTANT NOTE:

We perform an initial quality control of all revised manuscripts before re-review. Your manuscript will FAIL this control and the handling will be delayed IN CASE the following APPLIES:

- 1) A data availability section providing access to data deposited in public databases is missing. If you have not deposited any data, please add a sentence to the data availability section that explains that.
- 2) Your manuscript contains statistics and error bars based on $n=2$. Please use scatter blots in these cases. No statistics should be calculated if $n=2$.

When submitting your revised manuscript, please carefully review the instructions that follow below. Failure to include requested items will delay the evaluation of your revision.*****

- 1) a .docx formatted version of the manuscript text (including legends for main figures, EV figures and tables). Please make sure that the changes are highlighted to be clearly visible.
- 2) individual production quality figure files as .eps, .tif, .jpg (one file per figure). Please download our Figure Preparation Guidelines (figure preparation pdf) from our Author Guidelines pages <https://www.embopress.org/page/journal/14693178/authorguide> for more info on how to prepare your figures.
- 3) a .docx formatted letter INCLUDING the reviewers' reports and your detailed point-by-point responses to their comments. As part of the EMBO Press transparent editorial process, the point-by-point response is part of the Review Process File (RPF), which will be published alongside your paper.
- 4) a complete author checklist, which you can download from our author guidelines (<<https://www.embopress.org/page/journal/14693178/authorguide>>). Please insert information in the checklist that is also reflected in the manuscript. The completed author checklist will also be part of the RPF.
- 5) Please note that all corresponding authors are required to supply an ORCID ID for their name upon submission of a revised manuscript (<<https://orcid.org/>>). Please find instructions on how to link your ORCID ID to your account in our manuscript tracking system in our Author guidelines (<<https://www.embopress.org/page/journal/14693178/authorguide#authorshipguidelines>>)
- 6) We replaced Supplementary Information with Expanded View (EV) Figures and Tables that are collapsible/expandable online. A maximum of 5 EV Figures can be typeset. EV Figures should be cited as 'Figure EV1, Figure EV2' etc... in the text and their respective legends should be included in the main text after the legends of regular figures.

7) Please note that a Data Availability section at the end of Materials and Methods is now mandatory. In case you have no data that requires deposition in a public database, please state so instead of refereeing to the database.

See also <<https://www.embopress.org/page/journal/14693178/authorguide#dataavailability>>. Please note that the Data Availability Section is restricted to new primary data that are part of this study.

Additional information on source data and instruction on how to label the files are available

<<https://www.embopress.org/page/journal/14693178/authorguide#sourcedata>>.

10) Figure legends and data quantification:

- the name of the statistical test used to generate error bars and P values,
- the number (n) of independent experiments (please specify technical or biological replicates) underlying each data point,
- the nature of the bars and error bars (s.d., s.e.m.)

- If the data are obtained from n {less than or equal to} 5, show the individual data points in addition to the SD or SEM.

- If the data are obtained from n {less than or equal to} 2, use scatter blots showing the individual data points.

11) Our journal encourages inclusion of *data citations in the reference list* to directly cite datasets that were re-used and obtained from public databases. Data citations in the article text are distinct from normal bibliographical citations and should directly link to the database records from which the data can be accessed. In the main text, data citations are formatted as follows: "Data ref: Smith et al, 2001" or "Data ref: NCBI Sequence Read Archive PRJNA342805, 2017". In the Reference list, data citations must be labeled with "[DATASET]". A data reference must provide the database name, accession number/identifiers and a resolvable link to the landing page from which the data can be accessed at the end of the reference. Further instructions are available at <<https://www.embopress.org/page/journal/14693178/authorguide#referencesformat>>.

12) All Materials and Methods need to be described in the main text. We would encourage you to use 'Structured Methods', our new Methods format. According to this format, the Methods section should include a Reagents and Tools Table (listing key reagents, experimental models, software and relevant equipment and including their sources and relevant identifiers) followed by a Methods and Protocols section in which we encourage the authors to describe their methods using a step-by-step protocol format with bullet points, to facilitate the adoption of the methodologies across labs. More information on how to adhere to this format as well as downloadable templates (.doc or .xls) for the Reagents and Tools Table can be found in our author guidelines: <<https://www.embopress.org/page/journal/14693178/authorguide#manuscriptpreparation>>.

<<https://www.embopress.org/doi/10.15252/msb.20178071>>.

13) As part of the EMBO publication's Transparent Editorial Process, EMBO Reports publishes online a Review Process File to accompany accepted manuscripts. This File will be published in conjunction with your paper and will include the referee reports,

your point-by-point response and all pertinent correspondence relating to the manuscript.

I look forward to seeing a revised form of your manuscript when it is ready.

Kind regards,

Martina

Point by point responses:

Referee #1:

The authors provide evidence - using cell organoids - that E-cadherin-mediated cell sorting enables robust tissue patterning in the presence of noisy morphogen gradients.

The observation that cadherin-mediated cell sorting plays a decisive role in the process where morphogen gradients are converted into distinct tissue domains is not entirely novel. However, a systematic analysis of this mechanism using a synthetic morphogen system in 3D spheroids has not yet been undertaken.

While the manuscript is well written and the experimental results are clearly presented, several questions remain as to the correct interpretation of the results.

1. E-cadherin has both a mechanical (adhesive) and signaling function. From the data presented, it is not clear how E-cadherin triggers the observed cell sorting - do cells display random motion and sort by differential adhesion, or does E-cadherin signaling affect other cellular properties such as motility, protrusion formation etc. This can be addressed by using truncated/signaling defective forms of E-cadherin and a more in-depth analysis of cell behavior during the sorting process.

We thank the reviewer for raising important questions on mechanisms of how E-cadherin triggers the observed cell sorting. We think that the cell sorting observed here is mediated by both differential adhesion and signaling function of E-cadherin. As shown in Figure 5B, cells display random motion inside the spheroid, suggesting that cells in the spheroids can be sorted by cell surface differential adhesion. In addition, we here found that the cells start mixing regardless of the E-cadherin expression level when they express E-cadherin over a threshold level (Figure 5A and EV4). This result cannot be explained only with differential adhesion of cell surface and suggests that E-cadherin intracellular signaling involves the cell sorting.

To directly test if E-cadherin intracellular signaling is involved in the cell sorting in our L929-based system, we have done a cell sorting assay by mixing cells that express the full length E-cadherin or the E-cadherin mutant (Ecad Δ Cyto-GFP) in which the E-cadherin intracellular domain is replaced with GFP (Figure for rebuttal letter A and B). This mutant retains the homophilic affinity, but it does not convert the extracellular binding into the intracellular signaling due to the

loss of the intracellular interaction with adaptor proteins such as α -catenin, β -catenin, and vinculin. The E-cadherin intracellular domain is required for tissue compaction to form a tight aggregate (Figure for rebuttal letter C top panels). First, we confirmed that the mixture of parental L929 cells and E-cadherin-expressing L929 cells showed a clear cell sorting with a core region of E-cadherin-expressing L929 cells and an outer layer of parental L929 cells (Figure for rebuttal letter C bottom panels). Also, the mixture of parental L929 cells and Ecad Δ Cyto-GFP-expressing L929 cells showed a similar cell sorting, but the boundary between them was rough. When we mixed E-cadherin-expressing L929 cells and Ecad Δ Cyto-GFP-expressing L929 cells, E-cadherin-expressing L929 cells formed a core region, suggesting that the full-length E-cadherin induces a stronger cell adhesion than Ecad Δ Cyto-GFP. These results mean that the E-cadherin intracellular signaling could significantly contribute to the cell sorting in our system. To test how E-cadherin intracellular signaling triggers the cell sorting, we compared the cell surface protrusion formation of parental L929 cells, E-cadherin-expressing cells and Ecad Δ Cyto-GFP-expressing cells. However, the cell membrane dynamics were apparently not changed (Figure for rebuttal letter D). Though the detailed mechanisms of how E-cadherin intracellular signaling regulates cell sorting are interesting, they are not the primary scope of our study. Therefore, we decided to modify the discussion starting from Line 417 on page 9 in the Discussion section to clarify that the cell sorting and pattern formation is driven by both differential adhesion and E-cadherin intracellular signaling. In future studies, we will address the detailed mechanisms to gain deeper insights into cadherin-based cell sorting.

Figure for referee with unpublished data and its description has been removed upon request by the authors.

2. A prominent feature of E-cadherin-mediated cell-cell adhesion is its mechanosensitivity. Is there any evidence of mechanosensitivity playing a role during the sorting process? This can be easily analyzed by monitoring Vinculin recruitment to adhesion sites of alpha-catenin unfolding using suitable antibodies.

We thank the reviewer's comment on the cadherin mechanosensitivity. Cadherin induces mechanosensitive increase of cell adhesion strength with tension-induced vinculin recruitment and α -catenin oligomerization at the cadherin intracellular domain (Yonemura et al., *Nat. Cell Biol.* 12, 533–542 (2010); Noordstra et al., *Dev. Cell* 58, 1748-1763 (2023)). As these mechanisms occur at the E-cadherin intracellular domain, the above cell sorting data with the truncated E-cadherin mutant also suggests that the mechanosensitivity of E-cadherin is required for the cell sorting. While monitoring the dynamics of α -catenin and vinculin during the pattern formation process could provide valuable insights on cell sorting mechanisms, the detailed mechanisms of cell sorting are not the primary scope of our study. As it will be certainly interesting to analyze and engineer E-cadherin intracellular signaling and its mechanosensitivity to control cell adhesion strength in future studies, we added the discussion at Line 420 on page 9 in the Discussion section and referred to Yonemura et al., *Nat. Cell Biol.* 12, 533–542 (2010); Noordstra et al., *Dev. Cell* 58, 1748-1763 (2023).

3. One notable weakness of the synthetic assay system used is that the default cell-cell adhesion (in the absence of GFP signaling) is unclear and thus the cause for differential adhesion unknown. It would be much nicer to use a dual adhesion system where cells can express both E-cadherin and N-cadherin and GFP signaling would change the ratio of their expression. I realize that this might constitute some more work, but it would considerably improve this study.

We thank this constructive comment on how to address the unclearness of the default cell-cell adhesion. We know that L929 cells do not express classic cadherins such as E, N, P-cadherin, but we do not have detailed information about what kind of cell adhesion molecules except classic cadherins are expressed in L929 cells. To check more details of default cell-cell adhesion of L929 cells used here, we obtained and analyzed the two RNAseq data of L929 cells from NCBI GEO database. We found that L929 cells express a significant expression of Pcdh1 and Pcdhgc3, which are protocadherins (Pcdhs) in the cadherin superfamily. Pcdhs are mainly known to express in neural cells to regulate the neural circuit formation (Peek et al., *Cell Mol Life Sci* 74, 4133-4157 (2017)). Though we assume that Pcdhs potentially contribute to the default cell-cell adhesion, they do not induce tissue compaction by lacking the binding to the actin cytoskeleton at cytoplasmic domains in contrast to classic cadherins (Chen et al., *Development* 140, 3297-3302 (2013)). These data suggest that the default cell-cell adhesion of L929 does not disturb E-cadherin-based adhesion. We added this information at Line 147 on page 3 in the Result section.

4. A critical prerequisite for cell sorting is cell motility (random mixing), which is particularly important for eliminating ectopically activated cells. It might be quite revealing to modulate the level of cell motility in the assay system to evaluate the minimal level of cell motility as a function of the E-cadherin gradient.

We thank the reviewer for this insightful comment. We agree that cell motility is a prerequisite for cell sorting. However, it is quite difficult to purely suppress only cell motility using chemical treatments or knock-down experiments without affecting cell adhesion and cytoskeletal change in the 3D spheroid. To circumvent this problem, we observed the pattern formation in 2D system to reduce the cell motility by attaching cells on the tissue culture-treated plate (Fig. EV5A). We plated GFP-secreting cells in the left side of the insert wall and imC cells or iEcad-mC cells in the right side. Then we removed the insert wall and observed how GFP diffuses and induces the pattern in the right region. The result showed that iEcad-mC cells formed a gradient pattern similar to that of imC cells and did not induce the distinct domain with sharp boundary (Fig. EV5B). This result means that cell motility and frequent cell rearrangement are

essential for creating the pattern which we observed. We added this result at Line 280 on page 6 in the Result section.

Figure EV5. Pattern formation assay in a 2D culture condition.

Figure EV5. Pattern formation assay in a 2D culture condition.

(A) Experimental setup of pattern formation assay in a 2D culture system using a two-well culture-insert. 1.6×10^4 GFP-secretor cells were plated in the left side of the insert,

while 8×10^3 GFP-receiver cells were plated in the right side. The insert was removed after the cells were incubated overnight. The interaction between two regions were observed by confocal microscopy after another 48 h. **(B)** The mCherry distributions of both imC and iEcad-mC cells showed a gradient pattern. GFP-secretor cells were localized to a narrow area on the left. GFP-receiver cells were localized to the area to the right of the GFP-secretor cells, depicted in green and cyan in the upper cartoon above the confocal images, respectively. Scale bar: 500 μm . Fluorescence intensity profiles (green line: GFP, red line: mCherry) are overlaid below.

Taken together, this is an interesting manuscript, which needs a little more work before it is conclusive.

Referee #2:

Recently, there has been considerable interest in development synthetic morphogen systems, and a number of studies have developed systems in which secreted GFP can serve as a morphogen to receiving cells. In these systems, the GFP is captured by an anchor molecule and then presented to the neighboring cell which contains a nanobody for GFP coupled with the synNotch system. In this manuscript, Mizuno et al. develop a synthetic system for tissue patterning by combining one such synthetic morphogen system with receiver cells which express E-Cadherin in response to receiving the GFP signal. The authors show that coupling these two systems produces sharper boundaries in the receiving cells, and that this system can be used to create patterns and to study the mechanisms of robust patterning. Overall the experiments are well described and carried out, however, there are a number of conceptual and technical issues with the study that I describe below:

Major points:

1. It is clear from examining the data in Figs 1-3, that the expression of the cadherin molecule and the subsequent cell rearrangements that compactify the receiving domain also modify the gradient of GFP. This is true in both the constitutively expressed cadherin (Fig 2) and the cadherin induced by GFP (Fig 3), although the effects do not appear the same in these two cases. This will in turn feedback on the expression of the mCherry fluorescent protein and E-cadherin, and so some of the enhanced sharpness may come from sharpening the gradient, not only from cell rearrangements.

Thank you for your careful reading and raising an interesting possibility. We agree that E-cadherin-expressing receiver cells (imC^{Ecad} cells) formed a sharper gradient in Fig. 2C compared to the gradient in Fig. 1C. In our synthetic morphogen system, the gradient signaling range is determined by the balance between GFP production by secretor cells and GFP consumption by receiver cells (Toda et al., *Science* 370: 327-331 (2020)). Since imC^{Ecad} cells were more packed than normal receiver cells, the gradient range in the compact imC^{Ecad} spheroid became shorter than that in the loose imC spheroid. Here, E-cadherin expression did not change the responsiveness of receiver cells to various concentrations of GFP (Fig. EV3C, EV3D), suggesting that E-cadherin-expressing receiver cells respond to GFP just like normal receiver cells in the compact spheroid. This result suggests that E-cadherin induction will lead to the gradient sharpening by spheroid compaction but not to a feedback effect to the cells. The gradient sharpening would affect the width of uniformly-activated domain but not the interpretation process from a gradient input to switch-like pattern. To avoid the confusion raised by the reviewer, we added this point at Line 159 on page 4 in the Result section.

2. This effect is not captured in the mathematical model which uses a simple exponential to describe the GFP gradient. Thus, the model will not be able to accurately capture the different gradient shapes and relate the sharpness of the gradient to that of the induced pattern.

Please note that we have not obtained any experimental evidence supporting the explicit regulation of basic processes involved in creating the GFP gradient, such as decay and source level, in an E-cadherin dependent manner. Initially, we tried to incorporate the compaction effect into the mathematical model, but we consequently opted for a simpler condition to emphasize the main focus of this study, i.e., the DAE saturation regime. We believe that this strategy works well by avoiding unnecessarily complex regulatory systems. Utilizing an exponential form results from a basic setting involving simple diffusion of molecules with decay from a source, which is a standard approach for the morphogen model. Additionally, we would like to stress that the details of GFP profiles are not critical; rather, the emphasis should be placed on the DAE saturation regime for

this study. That said, we appreciate the reviewer's comment, and we acknowledge that exploring the function from E-cadherin expression level to GFP profiles using our synthetic engineering tools could be a potential extension of this study.

3. The authors do not consider the effects of cell division. It may be that much of the GFP is trapped upon entering the receiving cells but then spreads through cell division creating a gradient. Similarly, mCherry may be induced along the border and then the responsive domain may spread through growth rather than signal reception. This latter point could be addressed using a Notch inhibitor like DAPT to block signal reception at different times as the authors did in their previous study (Toda et al Science 2018). The effects of division are also not captured in the modeling but may play an important role in shaping the results.

Thank you for your insightful comment and suggestion. Cell division is one of the important factors that affect the multicellular pattern formation. As you suggested, it would be possible to use DAPT to observe the spread of mCherry positive regions by growth, but mCherry intensity would be diluted and become dim each time cell divide in this assay. To directly reduce the cell division in our system, we tested the lower serum concentration to slow down the cell division rate during the pattern formation (Fig. EV2). In our study, we normally cultured cells in the medium containing 10% FBS, in which the spheroid size was extended by cell division during 48 h culture. However, when we cultured cells in the medium containing 0.5% FBS, cell division rate was significantly reduced, leading to a smaller size of the spheroid at 48 h (Fig. EV2B). However, even in the 0.5% FBS condition, the uniformly-activated domain was similarly formed (Fig. EV2A). From this result, we conclude that cell division does not critically affect the overall pattern formation process, though cell division could trigger the spread of the GFP and responsive domain. The reduction of FBS could change not only cell division but many processes such as metabolism, but this result also shows the robustness of the pattern formation against variable serum concentration. We added this result at Line 198 on page 4 in the Result section.

Figure EV2. Robust pattern formation in a low serum condition.

Figure EV2. Robust pattern formation in a low serum condition.

(A) Confocal images of iEcad-mC spheroids incubated in DMEM containing 0.5% FBS (left) or 10% FBS (right). GFP-secretor spheroids with 50 GFP-secretor cells and GFP-receiver spheroids with 200 imC or iEcad-mC cells were separately formed and co-cultured from time 0 h. Spheroid images acquired at 0 and 48 h using a confocal microscope. GFP secretor cells express nucleus-localized IFP2.0, shown as a yellow pseudocolor. Scale bar: 100 μm . mCherry distributions are visualized by 16 pseudocolors in the bottom images. Fluorescence intensity profiles within rectangular regions (green line: GFP, red

line: mCherry) are overlaid at the bottom of the images. Shaded areas in the graph represent SD. 0.5% FBS: n=8, 10% FBS: n=6. **(B)** The graph indicated the spheroid size at 0 and 48 h when incubated with 0.5% FBS (white box) and 10% FBS (gray box). The spheroid size slightly increased in a 0.5% FBS condition after 48 hours of incubation, whereas it increased more than double in a 10% FBS concentration. Data are presented as mean \pm SD. ***P < 0.001, as determined using Welch's t test. 0.5% FBS: n=8, 10% FBS: n=6.

4. The quantifications and statistics are insufficient throughout. The methods indicate that the authors simply manually take a line profile through the aggregate using ImageJ. In each case, it appears that a single such profile is presented. The authors should quantify several aggregates for each case using automated image analysis rather than manual line profiles. The results should be presented with averages over aggregates as well as error bars and differences between conditions (i.e. with cadherin and without) should be evaluated statistically.

Thank you for your constructive comment. When we quantified the mCherry distribution along the GFP gradient axis in all the replicate spheroids (Fig. 1C-D, 2C-D, 3B-C, 3E, EV2A, and Appendix Fig. S3C, S4B-C), we manually set the rectangular ROI around the center of GFP-receiving spheroids. As suggested by the reviewer, we should have avoided any manual processes in the quantification methods. We have developed a following automated method.

1. Calculate the centroid positions of GFP-secreting cells using a binarized IFP channel
2. Automatically set the rectangular ROI from the centroid position to the end of a receiver spheroid.
3. Quantify the mCherry profile along the GFP gradient axis
4. Calculate the Hill coefficient, followed by statistical analysis.

Using this new automated method, the differences in the mCherry distribution between each receiver type were shown in Figure 3E and Appendix Figure S3. We also added the averaged data of mCherry distribution with replicates at 48 h in Fig. 1D, 2D, 3C with shaded areas representing SD. We have revised the methods at Line 605 on page 13 in the Materials and Methods section.

5. The mathematical model should be carefully compared to the data. Without doing this, the model doesn't add much to the paper.

Thank you for your constructive comment. We would like to clarify that the reviewer might have missed some critical descriptions of the mathematical model regarding the comparison of settings and outputs with experimental data. Firstly, the function of GFP level for the expression level of E-cadherin was quantitatively matched with experimental data, as shown in Appendix Fig. S5A. For the simulation outputs, we demonstrated that the E-cadherin domain size, resulting from the fold change in GFP source level, was compared with the experimental data, as shown in Appendix Fig. S5C. For a more explicit description of quantitative comparison, we have revised the main text as follows: ‘Additionally, the model can quantitatively reproduce the experimental data regarding tunability of E-cadherin domain size (Appendix Fig. S5C).’ at Line 318 on page 7. Having said that, we acknowledge the limitations in quantitatively comparing cell dynamics and mechanical aspects due to measurement difficulties in the current setting, although we believe that comprehensive understanding will be achieved in a future study. Recognizing the point raised by the reviewer, we have added the following description to the Discussion at Line 423 on page 9: ‘A further study is required to characterize the details of the cytoskeletal and signaling features as well as incorporating additional ingredients, such as cell morphology, and mechano-chemical effects to the mathematical model’. We believe that your concerns have been addressed.

Appendix Figure S5. Parameter and cell tracing analysis in the mathematical modeling.

Appendix Fig. S5. Simulation analysis of the mathematical model.

(A) The correspondence curve of GFP concentration and E-cadherin-mCherry

induction level in iEcad-mC cells. The red dots are plotted according to the measurement in Fig. 4C. The blue line is a fitting curve used for the simulations of the mathematical model. (C) Morphogen-induced cell adhesion achieves tunable tissue domain formation. The red dots are plotted according to the measurement in Appendix Fig. S4. The blue dots show the domain sizes with variable amounts of GFP source level in the mathematical modeling. $n=10$.

6. The advance of the paper appears to be limited. My main expertise is in developmental biology and nothing truly new about developmental systems has been learned. Natural systems that use the principles here have already been described and are cited by the authors. I am not a synthetic biologist, however, the advance from this perspective appears to be limited as well. Both the systems using GFP as a synthetic morphogen and placing cadherins downstream of synNotch have been described previously, including by these same authors. The main new aspect of the paper is coupling these existing systems.

Thank you for your comment on the significance of our study. Though the concept combining morphogen signaling and cell adhesion has already been proposed from model animal studies, we believe that our constructive approach is still important to prove the concept and explore possible mechanisms that are overlooked due to the complexity of natural systems. Using model animal embryos, there have been many proposed mechanisms on morphogen-based patterning such as gene regulatory networks, cell adhesion, cell proliferation and cell competition. Here, our aim is not adding a new pattern formation mechanism in this list. We aim to test minimum sufficient mechanisms among them by reconstituting morphogen-based cell-cell communication with living cells. First, as previously reported, we expect that coupling morphogen signaling and cell adhesion could remove ectopically-activated cells to form a precise gradient. However, unexpectedly, we found that coupling of morphogen signaling and cell adhesion is sufficient to convert the gradient signaling into distinct active and inactive domains. Our reconstitution system allows us to fine tune E-cadherin expression level, leading to the finding of non-linear relationship between E-cadherin expression level and function, which is key to form a homogeneous domain under gradient signaling. We believe that our constructive approach provides a significant advance on mechanistic understanding of multicellular pattern formation.

Minor points:

1. It isn't clear how well this system will translate to other cell types, particularly epithelial cells which may already express cadherin molecules. The authors state in lines 325-327 that these are potential applications without explaining how this would work.

We thank the reviewer for the comment on the possible applications of our system. We agree with the reviewer's concern about how to apply the system into epithelial cells that already express cadherin molecules. We have discussed this point in the discussion section: "In epithelial cells, intrinsic junctions with neighboring cells and the basal membrane make it impractical to organize multicellular patterns by inducing differential cadherin expression. Instead, other mechanisms are required for robust patterning of epithelial tissues, such as cell competition to eliminate EACs within a morphogen gradient. Introducing the SYMPLE3D concept into epithelial cells would be interesting to screen for responsive gene induction in response to morphogens that can generate precise tissue patterns.". However, it is confusing how to apply the SYMPLE3D system which is based on 3D cell aggregate into 2D epithelial tissues. To avoid the possible confusion raised by the reviewer, we removed the discussion related to epithelial cells around Line 400 on page 9 in the Discussion section.

2. In figures 1C, 2C, 3B, the authors should find a different way to label the GFP-secretors (IFP2.0 label); at first glance, it appears to show high mCherry intensity in secretor cells.

We thank the reviewer for pointing out the inappropriate use of pseudocolors. We removed the fluorescence of the secretor cells to avoid the confusion in Fig. 1C, 2C and 3B.

Referee #3:

This is an interesting work using synthetic biology to model tissue patterning. The authors found the coupling of a GFP-based synthetic morphogen gradient with E-

cadherin-based cell adhesion can improve the robustness of tissue patterning. The crosstalk between morphogen gradient and E-cadherin revealed a surprising switch-like effect. The cells exhibit compaction when the E-cadherin level crosses a critical threshold. This switch-like behavior led to a sharp boundary between two domains with uniformly high E-cadherin expression and low E-cadherin expression.

The principles revealed by this work are interesting to the general audience of developmental biologists, adhesion cell biologists, and synthetic biologists. The idea that coupling morphogen signaling and cell adhesion can enhance precision in tissue patterning is well-established and not new. The SYMPLE3D platform offered a new synthetic system that confirms this established concept. The artificial and tunable components of SYMPLE3D will be useful for future studies aiming to dissect various crosstalk mechanisms between morphogen signaling and adhesion. In my opinion, the most significant conceptual advance is the idea that a switch-like behavior of cell compaction in morphogen-driven cadherin expression can generate two spatial domains with a sharp boundary.

I am generally positive about this work. The experiments are well done and well-explained in the manuscript. I only have some suggestions for clarification and extra information that might be beneficial for readers.

What cadherins do L929 cells express? It appears that L929 cells have some basal adhesiveness. For example, the P-cadherin expressing GFP secretor forms significant adhesion with imC cells without the manipulation of E-cadherin. What adhesion molecules hold these cells together? A more thorough analysis of cadherins expressed in the L929 cells would help interpretation of later results (e.g., the threshold of E-cadherin over-expression)

Thank you for the comment on the basal adhesiveness of L929 cells. We know that L929 cells do not express classic cadherins such as E, N, P-cadherin, but we do not have detailed information about what kind of cell adhesion molecules except classic cadherins are expressed in L929 cells. To check more details of default cell-cell adhesion of L929 cells used here, we obtained and analyzed the two RNAseq data of L929 cells from NCBI GEO database. We found that L929 cells express a significant expression of *Pcdh1* and *Pcdhgc3*, which are protocadherins (Pcdhs) in the cadherin superfamily. Pcdhs are mainly known to express in neural cells to regulate the neural circuit formation (Peek et al., *Cell Mol Life Sci* 74, 4133-4157 (2017)). Though we assume that Pcdhs potentially contribute to the default cell-cell adhesion, they do not

induce tissue compaction by lacking the binding to the actin cytoskeleton at cytoplasmic domains in contrast to classic cadherins (Chen et al., *Development* 140, 3297-3302 (2013)). These data suggest that the default cell-cell adhesion of L929 does not disturb E-cadherin-based adhesion. We added this information into the result section at Line 147 on page 3.

Choice of P-cadherin & E-cadherin: Why did the authors choose to over-express P-cadherin in GFP secretors and E-cadherin in imC cells? Is there a specific logic behind it? As many cadherins form heterophilic adhesion, are P-cadherin and E-cadherin chosen because they do not form heterophilic adhesion? Are there previous work to support this? If there are, citations would be needed. There are likely more, but one related work is Min Bao, ..., Magdalena Zernicka-Goetz *Nature Cell Biology* 2022, where P-cadherin and E-cadherin are utilized to improve sorting of stem cell derived germ layer progenitors.

We thank the reviewer's comment on the choice of P-cadherin and E-cadherin in this paper. The reason that we used this combination is that the secretor and receiver cells could be clearly segregated to form organizer and receive aggregates, as referenced in the following study (Glykofrydis et al., *ACS Synth. Biol.* 10, 1465–1480 (2021)). Also, as the reviewer commented, the study by the Magdalena group demonstrated that differential cadherin expression between E-cadherin and P-cadherin drives initial cell sorting for precise blastocyst formation (Bao et al., *Nat Cell Biol* 24, 1341–1349 (2022)), which is related to our work. We have added the sentence "The secretor cells expressing P-cadherin and imC^{Ecad} cells were clearly segregated by heterophilic adhesion" at Line 155 on page 4 and cited the above papers.

Switch-like compaction on induced E-cadherin: Figures 4 and 5 are very interesting, and potentially challenge the classic differential adhesion hypothesis (DAH). The DAH suggests cells can sort based on quantitative differences of cadherins. Here the result suggests the sorting behavior is "all-or-none". The cells either express more E-cadherin than the threshold and mix well with the other E-cad-high cells or vice versa. I would love to see more conditions in Fig 5A. What range of E-cadherin expression can E-cad-high cells mix well? (10-fold-difference? 100-fold-difference?) Can the author (1) estimate how much difference in E-cad level between Ecad-high cells vs. iEcad-mC

cells in Fig 5A and (2) test how much difference in E-cad level can cells still sort to the same E-cad-high compartment? Is there no difference or will there be a bias (say E-cad-super-high cells still cluster together more than E-cad-medium-high cells) when the fold-difference is higher?

We thank the reviewer's comment on the E-cadherin expression range that can mix well. To estimate how much difference in E-cadherin expression levels between Ecad-high cells and iEcad-mC can be tolerated to induce cell mixing, we measured their E-cadherin expression levels with immunostaining using an anti-E-cadherin antibody (Fig. EV4D). First, our results showed that the E-cadherin induction levels in iEcad-mC cells increases proportionally with the increase of GFP concentration. At the concentration 0.0625 nM GFP, where iEcad-mC cells started mixing with Ecad-high cells (Fig. EV4B), there was approximately a 35-fold difference between Ecad-high and iEcad-mC cells. These results indicated that approximately a 35-fold difference between two cell types can be tolerated to induce the cell mixing behavior. When the fold-difference is higher by stimulating cells with lower concentration of GFP (0.03125 nM GFP), most iEcad-mC cells attached to the surface of the Ecad-high cells' spheroid and did not show mixing (Fig. EV4B), suggesting that the E-cadherin expression range should be less than 35-fold to induce cell mixing. We added this data into Fig. EV4D and described this result at Line 266 on page 6 in the Result section.

Figure EV4. Mixing of E-cadherin high-expressing cells and iEcad-mC cells.

(D) E-cadherin expression levels of parental L929 cells, Ecad-high cells, and iEcad-mC cells in response to varying GFP concentrations were analyzed with immunostaining using an anti-E-cadherin antibody. The mean value difference between Ecad-high and

iEcad-mC cells at 0.0625 nM GFP (dashed lines) is around 35-fold.

Mathematical model: The mathematical model aims to capture the sorting dynamics under GFP-induced E-cadherin gradient. While no mathematical model is perfect, I do believe this model is on the simpler side, almost too simple that intuition (like Fig 7) can already predict the simulation outcome. The author set up a relatively artificial scheme, where there is zero differential adhesion energy when E-cadherin in both neighbors exceeds a threshold. The y direction follows a periodic boundary condition (so no change of spheroid shape). The E-cadherin gradient naturally follows the quintic function and the sorting outcome is expected. This model does not consider contact size fluctuations (dynamic increase or shrink of cell contacts) and the complex neighbor exchange dynamics as it assumes the edge length is constant and the cells are simply swapping positions. By this simplification, the model becomes confirmatory of existing intuition, rather than generating new insights. The added value of Fig 6 is therefore not too different from a conceptual cartoon (Fig 7).

Firstly, we must emphasize the importance of the effective saturation regime for the E-cadherin-based DAE. This is not an artificial scheme but a rational interpretation based on experimental findings from this study. Incorporating this assumption into the model allows for the creation of clearer boundaries compared with the non-saturation regime, which is the main focus of our study. Secondly, while it is true that the E-cadherin gradient naturally follows a quintic function and sorting is expected, this alone is insufficient to achieve the steeper E-cadherin gradient observed (Fig. 6D-D'). Please note that the primary issue is not the cell sorting, but boundary formation.

Additionally, the periodic y-axis condition is not an artificial setting but a reasonable implementation, as the spheroid tissue does not have a physical end along the y-axis. To address the reviewer's concerns, we examined the effect of a no-flux boundary condition along the y-axis and found no significant difference compared to the periodic boundary condition. The results below are provided for the reviewer's reference: Figures A and A' below were obtained under periodic boundary conditions (BC), and Figures B and B' were obtained under no-flux BC.

Once again, implementing periodic boundary conditions is a natural approach in numerical simulations of this kind. Therefore, we decided to present the results with the periodic boundary condition along the y-axis.

Lastly, we emphasize that the mathematical analysis not only confirms intuition (although most theoretical results become intuitive in the end) but also provides insights into how sharp boundaries can be created. We demonstrated that two factors—1) cell mixing behavior within the DAE saturated domain and 2) a moderate E-cadherin turnover rate—are key to creating sharp boundaries. These arguments were initially described in the supplementary texts of the previous manuscript but have now been explicitly detailed in the main text (Fig. 6F-6J). In addition, we have included a new insight gained through theoretical analysis, which is difficult to obtain experimentally, thanks to your last comment (see below).

As a suggestion, a Cellular Potts Model (e.g., CompuCell3D) can more accurately capture cell contact dynamics and the evolution of domain shape (although the model has its inaccuracies too). Given that the experimental part is quite strong, I would not insist the author come up with a more sophisticated modeling scheme than the current version (But they should in future works!) Instead, the author should acknowledge all the limitations of the model (e.g., it is not modeling the whole cell but just considering the probability of cell switching position based on an artificial neighbor energy difference, it does not consider edge length fluctuation, the boundary condition does not allow the spheroid or the E-cad high domain to change shape in Y, etc.)

Thank you for your suggestion. Based on our extensive practical experience using CPM and other cell-centered models, we understand that they can create visually appealing images; however, they also increase complexity due to parameters that cannot be experimentally determined. We hope the reviewer understands that the emphasis should be placed on cell mixing behaviors due to the DAE saturation regime for creating a clear boundary. To focus on the main point, we maintain that the simple version, which captures the essence of measurable quantities, is more suitable for the objective of this study. Nonetheless, we fully agree that we should develop an extended model to capture additional features, including contact area fluctuations, morphological changes of cells, and mechanical effects in future works. Therefore, we have added the following description to the Discussion at Line 423 on page 9: ‘A further study is required to characterize the details of the cytoskeletal and signaling features as well as incorporating additional ingredients, such as cell morphology, and mechano-chemical effects to the mathematical model.’

I do have one suggested addition to the math modeling. Does the author consider scenarios where the sorting threshold (θ) is fluctuating? As the model stands, all cells have identical thresholds to determine whether they sort or not. What if this θ is variable (which it most likely does)? How does this change the boundary between the E-cadherin high and low domains, and the absorption of EACs?

We appreciate your constructive suggestion. Based on this, we modeled the cell-to-cell variance in the saturation threshold for the DAE using a normal distribution with parameters mean θ and standard deviation σ . We then investigated how the maximum gradient of E-cadherin level, the index of boundary sharpness, changes with these parameters. As shown below, the maximum E-cadherin gradient value tends to decrease with an increase in cell-to-cell variation σ , indicating that greater heterogeneity in the saturation level results in a less sharp boundary. However, for changes in the threshold level θ , there is an intermediate level where the E-cadherin gradient reaches its maximum. When θ is extremely high, the saturation effect is abolished, resulting in a similar outcome to the no-saturation regime. When θ is extremely low, the DAE becomes ineffective throughout the entire region, causing all cells to undergo mixing without sorting, which ultimately reduces the maximum E-cadherin gradient. We have included this result in Appendix Fig. S7B. Thanks to your valuable feedback, we believe that the added value provided by the theoretical

analysis is now reinforced.

Dear Dr. Toda

Thank you for the submission of your revised manuscript to EMBO reports. I apologize for the delay in handling your manuscript, but we have now received the full set of referee reports that is copied below.

As you will see, all referees are very positive about the study and request only minor changes to clarify and discuss the data and conclusions on cell division in Figure EV2.

Browsing through the manuscript myself, I noticed a few editorial things that we need before we can proceed with the official acceptance of your study.

- The manuscript sections should be in the following order: Title page - Abstract & Keywords - Introduction - Results - Discussion - Methods - Data Availability - Acknowledgments - Disclosure Statement & Competing Interests - References - Figure Legends - (Main Tables with legends) - Expanded View Figure Legends.
- "Materials and Methods" should be "Methods"
- "Figures and Tables" should be "Figures" as there are no tables.
- Please provide up to 5 keywords.
- Data and materials availability should be renamed to 'Data Availability' and moved to the end of the Methods section. Please remove the following statement: 'All data are available in the main manuscript and supplementary materials'. This section should only refer to data deposited in a public repository. If you archive the code in the Zenodo repository, please insert a link that resolves directly to the code at Zenodo (in addition to the Github link).
- Please note that the database name and accession code are not provided separately in the data availability statement. The preferred nomenclature in the Data Availability section is given as an example below. Both, the database name and the accession code should be stated and the URL needs to resolve directly to the dataset.

Example/Outline:

- RNA-Seq data: Gene Expression Omnibus GSE46843 (<https://www.ncbi.nlm.nih.gov/geo/query/acc.cgi?acc=GSE46843>)
- [data type]: [name of the resource] [accession number/identifier/doi] ([URL or identifiers.org/DATABASE:ACCESSION])
- Please update the 'Conflict of interest' paragraph to our new 'Disclosure and competing interests statement'. For more information see <https://www.embopress.org/page/journal/14693178/authorguide#conflictsofinterest>
- Regarding the Author Contributions, we now use CRediT to specify the contributions of each author in the journal submission system. Therefore, please remove the Author Contributions from the manuscript file and make sure that the author contributions in our online manuscript tracking system are correct and up-to-date. The information you specified in the system will be automatically retrieved and typeset into the article. You can enter additional information in the free text box provided, if you wish.
- The Data References should be part of the main References list. The abbreviation/prefix in the manuscript text is 'Data ref:' instead of 'Data reference'.
- Please note that although dataset specific (<https://0-www-ncbi-nlm-nih-gov.brum.beds.ac.uk/geo/query/acc.cgi?acc=GSE202376>) URL is provided, the URL is currently invalid. This needs to be rectified.
- Please provide callouts to Figure panels 6F and 6G.
- Author Checklist: please add the missing information on corr. author's name, journal, and manuscript number (top left).
- The information on funding in the Acknowledgment section and in the online manuscript tracking system must match. The following two funders are missing in the tracking system: World Premier International Research Center Initiative and National University of Singapore (NUS). This needs to be rectified.
- Please remove the movie legends from the Appendix pdf. It is sufficient to supply them as README.text file, as you did.
- Our production/data editors have asked you to clarify several points in the figure legends (see below). Please incorporate these changes in the manuscript and return the revised file with tracked changes with your final manuscript submission.

B) Statistical test information. Only p-values that are actually shown in the figure panel(s) should (and must) be defined in the legends, all others should be removed from (or added to) the legend. Moreover, we ask for the specification of exact p-values:
- Please note that the exact p values are not provided in the legends of figures 3e; EV 2b; S4C.

B) Replicates and error bars:

- Please note that the box plots need to be defined in terms of minima, maxima, centre, bounds of box and whiskers, and percentile in the legends of figures 3e; EV 2b.
 - Although 'n' is provided, please describe the nature of entity for 'n' in the legend of figure EV 2b.
 - Please note that the measure of center for the error bars needs to be defined in the legends of figures 1d; 2d; 3c; 6e", i', j; EV 2a; EV 3c-d.
 - Please note that the scale bar needs to be defined for figures 2a-b; 3c; 5b; EV 1c.
- We routinely perform a figure check for all manuscripts prior to publication. In this case we noticed that the image for L929 shown in Figure 4C seems to have been re-used in Figure EV3A. Please clarify whether this control applies to both experiments, i.e., whether the data shown in Figure 4C and in Figure EV3A are from the same experiment. If so, please either state the re-use clearly in the figure legend or - preferred - provide different control samples/images to show reproducibility.
- Similar observations were made for Figure EV1C and Appendix Fig S3B: the GFP-secretor image appears to be the same.
 - The bright field series in Figure Figure EV3B and Appendix Fig S2A seems to be same, but EV3B shows mCherry and S2A GFP. Please check whether these images are from the same experiment and displayed twice and if so, please clearly state so in the figure legend to avoid any ambiguities.
 - Please respond to the above queries in a point-by-point response to clarify.
 - Please upload the source data as one folder per figure.
 - Please describe your findings in the Abstract in present tense.

- On a different note, I would like to alert you that EMBO Press offers a new format for a video-synopsis of work published with us, which essentially is a short, author-generated film explaining the core findings in hand drawings, and, as we believe, can be very useful to increase visibility of the work. This has proven to offer a nice opportunity for exposure i.p. for the first author(s) of the study. Please see the following link for representative examples and their integration into the article web page:
https://www.embopress.org/video_synopses
<https://www.embopress.org/doi/full/10.15252/emj.2019103932>

Kind regards,

Referee #1:

The authors have adequately addressed my concerns. I appreciate their efforts in clarifying the baseline adhesion of the L929 cells, quantifying the threshold level of E-cadherin that enables the mixing of E-cadherin high cells, and investigating the effect of the varying threshold of E-cadherin in the formation of a sharp boundary. While I still believe the mathematical model can be improved, in particular incorporating realistic cell neighbor exchange beyond a simple swap on the grid, this is not critical for the points made by the authors. I will therefore recommend acceptance of the manuscript for publication at the present form.

Referee #2:

The manuscript has been revised along the lines suggested by the three referees. The revised version has considerably

improved and is now suitable for publication in EMBO Reports.

Referee #3:

The authors have responded appropriately to the previous round of review. They have added some interesting data and clarified the text in places. My comments on novelty in the previous version apply to this version as well, and I leave it to the editor to decide whether the manuscript is suitable for EMBO Reports.

I have one comment on the new data in Figure EV2. The authors suggest that the results show that the gradient is robust to cell division, however, to me it appears that gradient is shorter and sharper in low division conditions which suggests a potential role for cell division in shaping the gradient. This is interesting and if this is true, it would be worth the authors discussing it. I don't think any further experiments are needed.

Dear Dr. Toda

Thank you for the submission of your revised manuscript to EMBO reports. I apologize for the delay in handling your manuscript, but we have now received the full set of referee reports that is copied below.

As you will see, all referees are very positive about the study and request only minor changes to clarify and discuss the data and conclusions on cell division in Figure EV2.

Browsing through the manuscript myself, I noticed a few editorial things that we need before we can proceed with the official acceptance of your study.

- The manuscript sections should be in the following order: Title page - Abstract & Keywords - Introduction - Results - Discussion - Methods - Data Availability - Acknowledgments - Disclosure Statement & Competing Interests - References - Figure Legends - (Main Tables with legends) - Expanded View Figure Legends.

We corrected the order of the manuscript sections.

- "Materials and Methods" should be "Methods"

We changed the section name from "Materials and Methods" to "Methods."

- "Figures and Tables" should be "Figures" as there are no tables.

We changed the section name from "Figures and Tables" to "Figures."

- Please provide up to 5 keywords.

We provided 5 keywords: Pattern formation, Synthetic biology, Morphogen, Cell adhesion, Cadherin.

- Data and materials availability should be renamed to 'Data Availability' and moved to the end of the Methods section. Please remove the following statement: 'All data are available in the main manuscript and supplementary materials'. This section should only refer to data deposited in a public repository. If you archive the code in the Zenodo repository, please insert a link that resolves directly to the code at Zenodo (in addition to the Github link).

We renamed, edited and moved the 'Data Availability' section. We also added a link to the Zenodo repository.

- Please note that the database name and accession code are not provided separately in the data availability statement. The preferred nomenclature in the Data Availability section is given as an example below. Both, the database name and the accession code should be stated and the URL needs to resolve directly to the dataset.

Example/Outline:

We edited the Data Availability section with reference according to the example.

- Please update the 'Conflict of interest' paragraph to our new 'Disclosure and competing interests statement'. For more information see

<https://www.embopress.org/page/journal/14693178/authorguide#conflictsofinterest>

We updated "Conflict of interest" to "Disclosure and competing interests statement."

- Regarding the Author Contributions, we now use CRediT to specify the contributions of each author in the journal submission system. Therefore, please remove the Author Contributions from the manuscript file and make sure that the author contributions in our online manuscript tracking system are correct and up-to-date. The information you specified in the system will be automatically retrieved and typeset into the article. You can enter additional information in the free text box provided, if you wish.

We removed the author contributions section and confirmed the author contributions in the online manuscript tracking system.

- The Data References should be part of the main References list. The abbreviation/prefix in the manuscript text is 'Data ref:' instead of 'Data reference'.

We revised this point at line 155 in the manuscript.

- Please note that although dataset specific (<https://0-www-ncbi-nlm-nih-gov.brum.beds.ac.uk/geo/query/acc.cgi?acc=GSE202376>) URL is provided, the URL is currently invalid. This needs to be rectified.

We deleted this data reference because the dataset was unexpectedly removed from the NCBI database during the revision process, although it existed at the time of submission.

- Please provide callouts to Figure panels 6F and 6G.

We corrected the incorrect figure callouts at lines 336 and 338, changing Fig. 7F-7G' and Fig. 7G' to Fig. 6F-6G' and Fig. 6G'.

- Author Checklist: please add the missing information on corr. author's name, journal, and manuscript number (top left).

We added the information in the Author Checklist file.

- The information on funding in the Acknowledgment section and in the online manuscript tracking system must match. The following two funders are missing in the tracking system: World Premier International Research Center Initiative and National University of Singapore (NUS). This needs to be rectified.

We added "National University of Singapore" in the tracking system. "World Premier International Research Center Initiative" is a part of "Ministry of Education, Culture, Sports, Science and Technology (MEXT)" in the funder list, but we could not find "World Premier International Research Center Initiative" in the suggestion list. Therefore, we added it in the box for the Grant Reference Number.

- Please remove the movie legends from the Appendix pdf. It is sufficient to supply them as README.text file, as you did.

We removed the movie legends from the Appendix file.

- Our production/data editors have asked you to clarify several points in the figure legends (see below). Please incorporate these changes in the manuscript and return the revised file with tracked changes with your final manuscript submission.

B) Statistical test information. Only p-values that are actually shown in the figure panel(s) should (and must) be defined in the legends, all others should be removed from (or added to) the legend. Moreover, we ask for the specification of exact p-values:

- Please note that the exact p values are not provided in the legends of figures 3e; EV 2b; S4C.

We provided the exact p values in the legends of Fig. 3E, EV2B, S4C.

B) Replicates and error bars:

- Please note that the box plots need to be defined in terms of minima, maxima, centre,

bounds of box and whiskers, and percentile in the legends of figures 3e; EV 2b.

We defined and noted the box plots in the legends of Fig. 3E, EV2B, S4C.

- Although 'n' is provided, please describe the nature of entity for 'n' in the legend of figure EV 2b.

We described that “The same samples were observed and measured at 0 and 48 h in each condition (n=8 for 0.5% FBS, n=6 for 10% FBS).” in the legend of Fig. EV2B.

- Please note that the measure of center for the error bars needs to be defined in the legends of figures 1d; 2d; 3c; 6e", i', j; EV 2a; EV 3c-d.

We defined and noted the lines indicating means with error bars in the legends of figures Fig. 1D, 2D, 3C, 6E'-E", 6F-G, 6I-I', 6J.

- Please note that the scale bar needs to be defined for figures 2a-b; 3c; 5b; EV 1c.

We added the scale bar to the legends of Fig. 2A, 3D, 5B, EV1C, S2B, S3C, S4B.

- We routinely perform a figure check for all manuscripts prior to publication. In this case we noticed that the image for L929 shown in Figure 4C seems to have been re-used in Figure EV3A. Please clarify whether this control applies to both experiments, i.e., whether the data shown in Figure 4C and in Figure EV3A are from the same experiment. If so, please either state the re-use clearly in the figure legend or - preferred - provide different control samples/images to show reproducibility.

We replaced the image of L929 in Fig. EV3A with a different control image.

- Similar observations were made for Figure EV1C and Appendix Fig S3B: the GFP-secretor image appears to be the same.

We replaced the image of imC cells in Appendix Fig. S3B with an image from a different sample.

- The bright field series in Figure Figure EV3B and Appendix Fig S2A seems to be same, but EV3B shows mCherry and S2A GFP. Please check whether these images are from the same experiment and displayed twice and if so, please clearly state so in the figure legend to avoid any ambiguities.

We confirmed that these images are from the same experiment. We described that “These images are from the same samples as in Fig. EV3B. While Fig. EV3B shows bright-field and mCherry channels, this figure presents bright-field and GFP channels.” in the legend of

Appendix. Fig. S2A.

In addition to above comments, we found several errors in our manuscript, so we corrected as follows:

- We corrected the author affiliation.
- We corrected the figure callouts at lines 200 and 232.
- We corrected the catalog number of the plasmid at line 478.
- We corrected a reagent name at line 547.
- We inserted the figure numbers in the legend of Fig. 6 (at lines 1051 and 1052).
- In Fig.1C, we found that the same images were mistakenly used to show the spheroids at 12 h and 24 h during revision process. We replaced the spheroid images at 24 h in Fig. 1C with appropriate images.
- We inserted a space between the number and "h" in Fig. EV2B.
- We adjusted the character spacing of x-axis values in Fig. EV4D.
- We removed the small square points indicating the mean values from the box plots in Fig. 3E, EV2B, S4C, as they were not important.

- Please respond to the above queries in a point-by-point response to clarify.

- Please upload the source data as one folder per figure.

We uploaded the source data file as one folder per figure.

- Please describe your findings in the Abstract in present tense.

We edited the abstract.

- On a different note, I would like to alert you that EMBO Press offers a new format for a video-synopsis of work published with us, which essentially is a short, author-generated film explaining the core findings in hand drawings, and, as we believe, can be very useful to increase visibility of the work. This has proven to offer a nice opportunity for exposure i.p. for the first author(s) of the study. Please see the following link for representative examples and their integration into the article web page:

<https://www.embopress.org/doi/full/10.15252/embj.2019103932>

Please let me know, should you be interested to engage in commissioning a similar video

synopsis for your work. According operation instructions are available and intuitive.

Kind regards,

Referee #1:

The authors have adequately addressed my concerns. I appreciate their efforts in clarifying the baseline adhesion of the L929 cells, quantifying the threshold level of E-cadherin that enables the mixing of E-cadherin high cells, and investigating the effect of the varying threshold of E-cadherin in the formation of a sharp boundary. While I still believe the mathematical model can be improved, in particular incorporating realistic cell neighbor exchange beyond a simple swap on the grid, this is not critical for the points made by the authors. I will therefore recommend acceptance of the manuscript for publication at the present form.

We thank the referee for the positive feedback on our revision and for the recommendation of acceptance.

Referee #2:

The manuscript has been revised along the lines suggested by the three referees. The revised version has considerably improved and is now suitable for publication in EMBO Reports.

We thank the referee for the positive feedback on our revision and for the recommendation

of publication.

Referee #3:

The authors have responded appropriately to the previous round of review. They have added some interesting data and clarified the text in places. My comments on novelty in the previous version apply to this version as well, and I leave it to the editor to decide whether the manuscript is suitable for EMBO Reports.

I have one comment on the new data in Figure EV2. The authors suggest that the results show that the gradient is robust to cell division, however, to me it appears that gradient is shorter and sharper in low division conditions which suggests a potential role for cell division in shaping the gradient. This is interesting and if this is true, it would be worth the authors discussing it. I don't think any further experiments are needed.

We thank the referee for the positive feedback on our revision and the comment for the new data. As the referee mentioned, the relationship between cell division rate and boundary sharpness is interesting, but we think that our new data is not sufficient to conclude the relationship. Therefore, we added a following discussion at line 212 in the manuscript:

The boundary between active and inactive domains in low serum conditions may appear sharper than in normal conditions, but further studies are required to clarify how the cell division rate affects the boundary sharpness.

Dr. Satoshi Toda
Osaka University
Institute for Protein Research
3-2 Yamadaoka
Suita, Osaka 565-0871
Japan

Dear Dr. Toda,

I am very pleased to accept your manuscript for publication in the next available issue of EMBO reports. Thank you for your contribution to our journal.

Kind regards,
